# Genome wide analysis revealed conserved domains involved in the effector discrimination of bacterial type VI secretion system

Caihong Wang[1,6], Mingxing Chen[2,6], Yuhan Shao[1], Mengyuan Jiang[1], Quanjie Li[3], Lihong Chen[2], Yun Wu[1], Shan Cen [3,4], Nicholas R. Waterfield [5], Jian Yang [2✉] & Guowei Yang [1✉]

Type VI secretion systems (T6SSs) deliver effectors into target cells. Besides structural and effector proteins, many other proteins, such as adaptors, co-effectors and accessory proteins, are involved in this process. MIX domains can assist in the delivery of T6SS effectors when encoded as a stand-alone gene or fused at the N-terminal of the effector. However, whether there are other conserved domains exhibiting similar encoding forms to MIX in T6SS remains obscure. Here, we scanned publicly available bacterial genomes and established a database which include 130,825 T6SS *vgrG* loci from 45,041 bacterial genomes. Based on this database, we revealed six domain families encoded within *vgrG* loci, which are either fused at the C-terminus of VgrG/N-terminus of T6SS toxin or encoded by an independent gene. Among them, DUF2345 was further validated and shown to be indispensable for the T6SS effector delivery and LysM was confirmed to assist the interaction between VgrG and the corresponding effector. Together, our results implied that these widely distributed domain families with similar genetic configurations may be required for the T6SS effector recruitment process.

[1] Beijing Institute of Tropical Medicine, Beijing Friendship Hospital, Capital Medical University, Beijing 100050, China. [2] NHC Key Laboratory of Systems Biology of Pathogens, National Institute of Pathogen Biology, Chinese Academy of Medical Sciences & Peking Union Medical College, Beijing 102629, China. [3] Institute of Medicinal Biotechnology, Chinese Academy of Medical Sciences & Peking Union Medical College, Beijing 100050, China. [4] CAMS Key Laboratory of Antiviral Drug Research, Peking Union Medical College, Chinese Academy of Medical Sciences, Beijing 100730, China. [5] Warwick Medical School, Warwick University, Coventry CV4 7AL, UK. [6] These authors contributed equally: Caihong Wang, Mingxing Chen. ✉email: yangj@ipbcams.ac.cn; yangguowei@hotmail.com

The type VI secretion system (T6SS) is a molecular device that delivers effectors to kill neighboring prokaryotic and eukaryotic cells using a contractile injection mechanism[1–4]. T6SS effectors can be classified into "specialized" and "cargo" effectors depending on the modality of the delivery system[3,5–8]. In the case of a specialized effector, the toxin domain is fused to the structural components of T6SS complex, such as VgrG, Hcp or PAAR proteins[3,9,10]. In contrast, cargo effectors are associated with the structural components through non-covalent interactions.

Effector delivery of specialized secretion systems is important for many virulent phenotypes caused by pathogenic bacteria[11–13]. T6SSs deliver effectors into target cells with the help of some proteins, including adaptors, co-effectors, and etc. The adaptor proteins (sometimes referred to as chaperones) may have variable functions relating to the effectors, such as the proper folding/ unfolding, stability and recruitment to the secretion apparatus[14,15]. Similar to the well characterized chaperone/ adaptors in the type III secretion system (T3SS) and type IV secretion system (T4SS), three T6SS adaptor families have been characterized, including DUF4123 (Tec/Tap), DUF1795 (DcrB/ Eag) and DUF2169. These domains are sometimes encoded adjacent to *vgrG* and facilitate the interactions between the VgrG and the cognate effector specifically. These adaptor proteins are not secreted but are indispensable for the delivery of their cognate effectors[16–18].

Different from classic adaptors that are encoded as single proteins, several conserved domains were identified primarily located at the N-terminus of toxins in the T6SS *vgrG* neighborhood and can be used as markers for T6SS effectors, such as MIX (Marker for type sIX effectors) and FIX (Found in type sIX effector) domains[7,19]. The MIX domain, encoded by a single gene, can be secreted along with the cognate effector via the T6SS and has been termed as a T6SS co-effector. Fused at the N-terminus of the effector, MIX domain is also necessary for the secretion of the toxin[20,21]. Considering the diversity of well characterized T6SS effectors, it is possible that more conserved domains may be encoded that assist the effector delivery of T6SS in this complex secretion system.

T6SS propels its "tube-spike complex" through the membranes of target cells in order to deliver effector proteins. As central components of the spike complex, VgrG proteins fall into two classes, "structural" and "evolved" VgrGs. Evolved VgrG proteins are defined as polymorphic toxins with conserved N-terminal VgrG domains and variable C-terminal toxin domains (i.e., specialized effectors)[3,13,22]. Previous studies revealed that not only T6SS effector and immunity proteins but also many T6SS-related proteins, including the aforementioned adaptors and co-effectors, are frequently encoded in the *vgrG* neighborhood[20,23].

Here, we used a comprehensive bioinformatic approach to scan currently available bacterial genomes and created a VgrG database (http://www.mgc.ac.cn/dbVgrG/), which contains 130,825 VgrG protein sequences from 45,041 Gram-negative bacterial genomes. In-depth analysis of this dataset revealed that, besides the MIX domain, six domain families within the *vgrG* loci are not only encoded as independent genes, but also fused at the C-terminus of VgrG and/or the N-terminus of toxin, including DUF2345 (cl01733), FIX-like (cl41761), LysM (cl21525), 5 (cl33691), PG_binding_1 (cl38043) and PHA00368 (cl30808) domains. These families are widely distributed among diverse bacterial genomes and associated with dozens of different toxins. Further experiments confirmed that DUF2345 and LysM containing proteins assist the interaction between VgrG and the corresponding effector, which are potentially important in the discrimination process of T6SS effectors.

## Results

**Construction of the VgrG database.** Encoded as a stand-alone gene or fused at the N-terminus of the toxin, the MIX domains can assist the delivery of their cognate T6SS effector[19,20]. As the central component of the spike complex, VgrG is a good marker to explore the potential conserved domains involved in the delivery of T6SS effectors. Therefore, we set out to create a comprehensive dataset of VgrG proteins from available Gram-negative genome sequences lodged in the public GenBank database.

Previous studies have revealed that the Afp8 proteins of extracellular contractile injection systems (eCISs) are homologous to VgrG proteins, thus representing a potential confounding influence on the integrity of the dataset[24–26]. Therefore, we firstly downloaded 872 experimentally verified VgrG proteins from the established SecReT6 T6SS database[27]. It provides a positive control dataset to better avoid potential false positive hits (such as Afp8 homologs). A bioinformatic scan for conserved domains confirmed that the VgrG domain (accession: COG3501) was present in all 872 verified VgrG proteins in addition to 472 Afp8 proteins available from the dbeCIS database[26]. Importantly, the identified domains found in 861 (99%) verified VgrGs range between 451 and 750 amino acids, whereas there are only 10 (2%) Afp8 proteins that fall within this size range (Fig. 1a). We therefore proposed the use of an "empirical criterion" for the further systematic screening for bona fide VgrG proteins in the 133,722 publicly available bacterial genomes (Fig. 1a). Using this approach, a total of 130,825 VgrG proteins were successfully identified from 45,041 Gram-negative bacterial genomes.

To further characterize the VgrG proteins identified above, we constructed a maximum-likelihood (ML) phylogenetic tree based specifically on the sequences of the conserved VgrG domains (Fig. 1b). Using the aforementioned 872 previously defined VgrGs as indicators, we observed that our ML tree exhibited a similar overall topology regarding types/subtypes of T6SS operons as previously described[27], supporting the validity of our approach.

**Identification of conserved domains with multiple encoding forms within *vgrG* loci.** Firstly, a screen was performed to identify MIX containing protein, based on the aforementioned VgrG database. A total of 7208 MIX containing proteins within *vgrG* loci were identified, which are widely distributed among various bacteria (Supplementary Fig. 1). Importantly, sandwiched between *vgrG* and downstream effector gene, MIX domain exhibit multiple encoding configurations including single proteins and fusions at the C-terminus of VgrG or N-terminus of effector (Supplementary Fig. 2).

Based on the encoding features of MIX domain, we then developed a screening strategy to identify more conserved domains with similar multiple encoding configurations as MIX within *vgrG* loci from the VgrG database created above (Fig. 2). In brief, we scanned a maximum of three downstream genes of each *vgrG* locus to collect the conserved domains within the proteins sandwiched by *vgrG* and downstream toxin (if present). A domain family was reported if it was present in both of two encoded forms: "stand-alone" gene (i.e., single form) and fused to either the C-terminus of VgrG or the N-terminus of a toxin (i.e., fusion form). Finally, to further explore the presence of these domain families within *vgrG* loci in finer detail, we extended our search without the limitation of linkage to known toxins to identify more candidate domain-containing proteins within *vgrG* loci (Fig. 2).

After the screening process and careful manual curation, DUF2345 (cl01733), FIX-like (cl41761), LysM (cl21525), 5

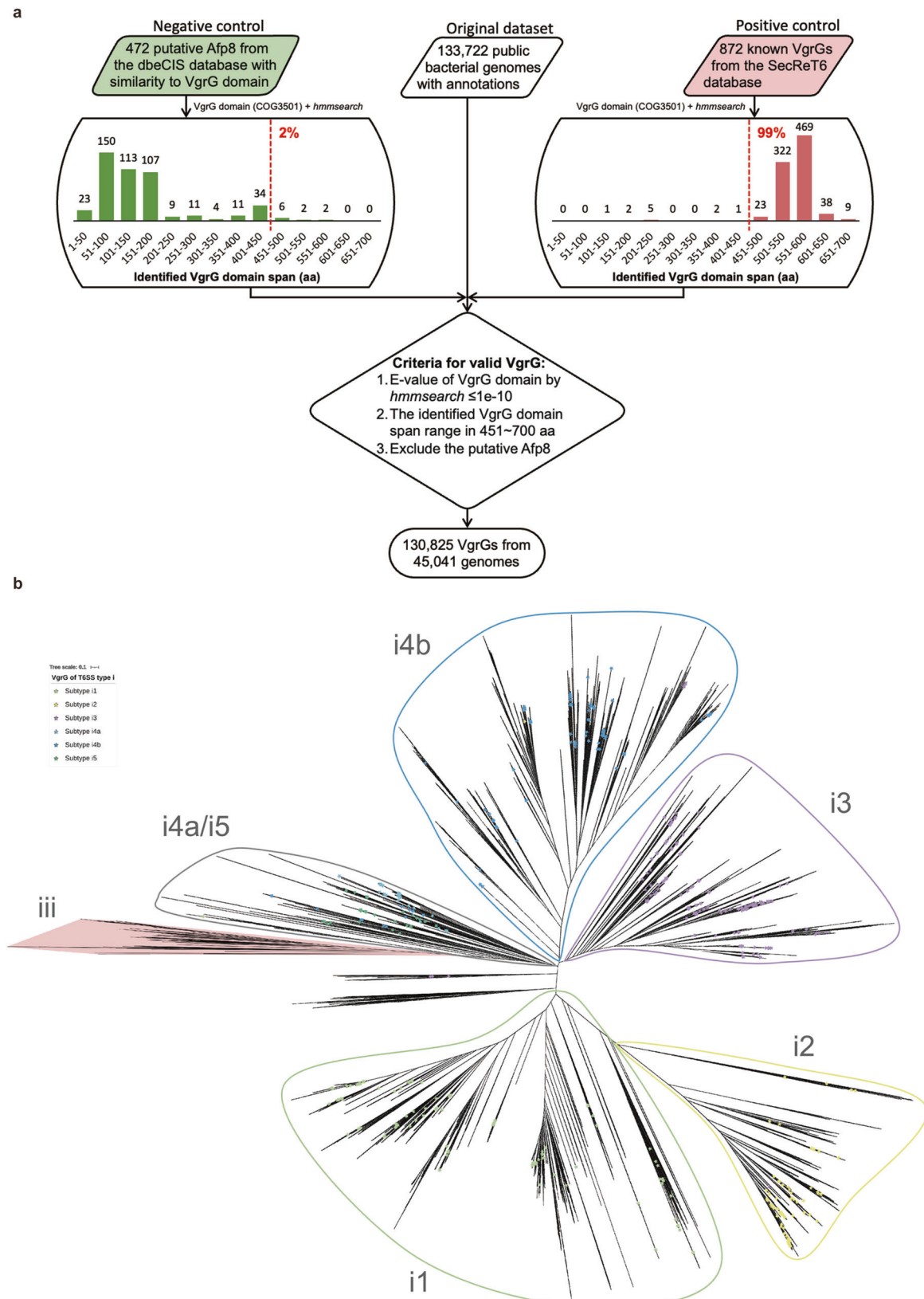

(cl33691), PG_binding_1 (cl38043) and PHA00368 (cl30808) were successfully identified (Supplementary Table 1). As shown in Supplementary Fig. 3, besides the single form, all these domain families have at least one fusion form. Further, the FIX-like (cl41761), LysM (cl21525), 5 (cl33691) and PG_binding_1 (cl38043) families can be found in both fusion forms. Notably,

some of them were encoded adjacent to known T6SS adaptor, which implies that their functions can be different from T6SS adaptors.

Besides MIX domain, three well characterized T6SS adaptor families (DUF4123, DUF2169, and DUF1795) had been reported to assist the interaction between VgrG and its cognate effectors.

**Fig. 1 VgrG identification workflow and an unrooted phylogenetic tree of VgrGs for the demarcation of T6SS subtypes. a** The workflow for the identification of valid VgrGs from 133,722 publicly available bacterial genomes. The 872 VgrGs available from the established T6SS database SecReT6 (red) and 472 putative Afp8 proteins, encoding VgrG domains, available from the eCIS database dbeCIS (green) were used as positive and negative datasets respectively for the selection of the empirical criteria for large-scale VgrG screening. **b** The 872 VgrGs available from the SecReT6 database with predefined subtype information are indicated by colored stars (key). VgrGs from subtypes i4a and i5 were mixed within the same clade in the tree, but these two subtypes were indeed closely related in the previous study[27]. The known type iii T6SS clade, derived mostly from *Bacteroidetes*, is highlighted with red shadow.

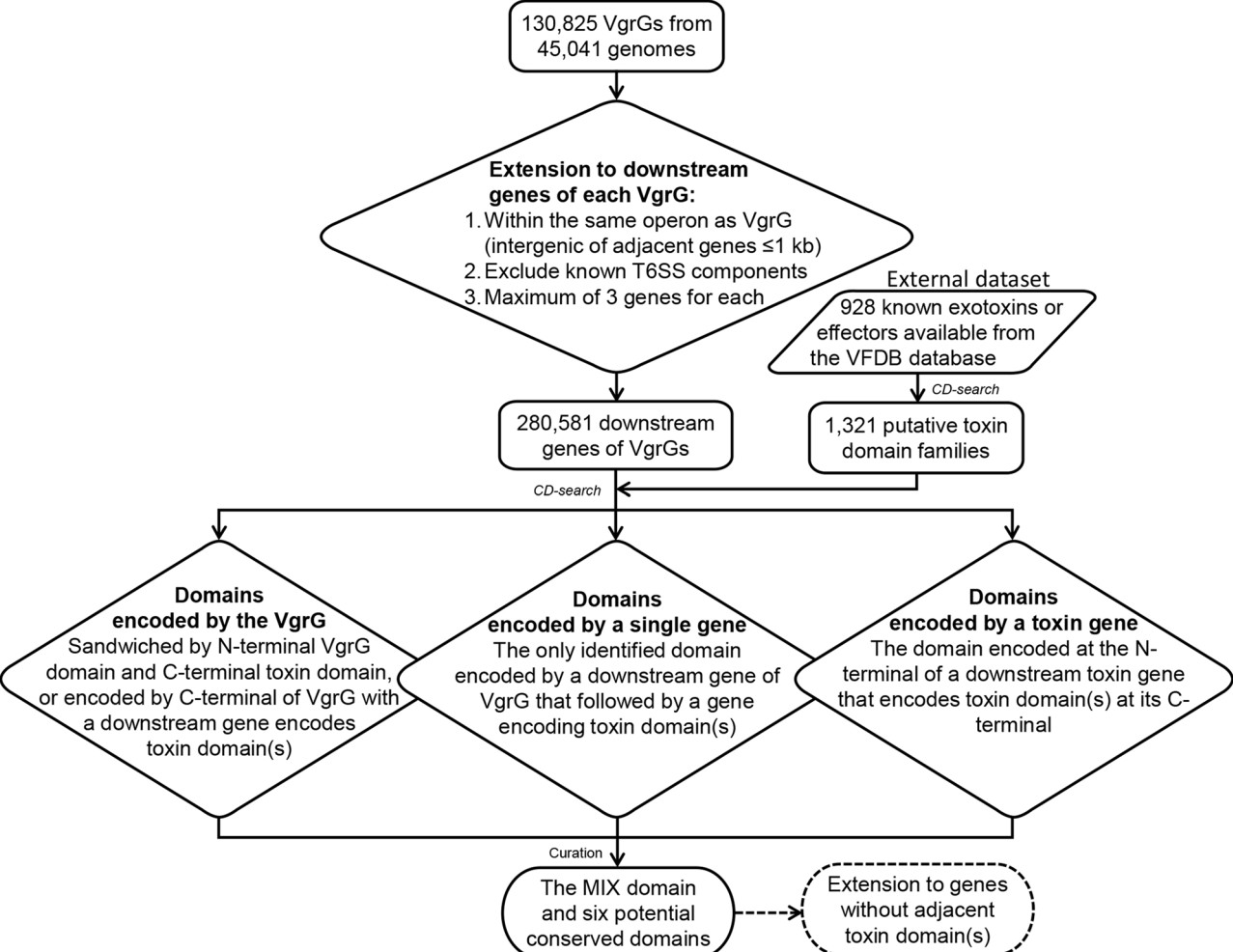

**Fig. 2 The detailed identification workflow for conserved domains with multiple encoding forms within *vgrG* loci.** For each *vgrG* locus, a maximum of three continuous downstream genes encoded on the same strand as *vgrG*, with an intergenic distance between adjacent genes of <1 kb were collected. Known components of the T6SS operon and any annotated pseudogenes were excluded. Then, the 280,581 remained downstream genes were scanned for conserved domains by batch CD-search. A total of 1321 putative toxin domain families were deduced from a collection of 928 experimentally verified exotoxins/effectors available from the VFDB database[53]. Each domain family identified within downstream genes dataset were further classified into three cases for final manual curation and determination.

We further screened these adaptor families encoded within *vgrG* loci. Amongst 130,825 *vgrG* loci, besides three adaptor domains (37.44%) and MIX domain (3.14%), 31.33% of *vgrG* loci encode at least one of the six conserved domain families identified here. In contrast, only 28.09% of *vgrG* loci do not include any of the adaptor/MIX/conserved domains mentioned above (Supplementary Fig. 4).

**DUF2345 domain assist the delivery of T6SS effector.** Although DUF2345 is considered as an extension of the VgrG gp5 domain, it is not encoded by all VgrGs[6,28,29]. Nevertheless, among the aforementioned six conserved domains, the DUF2345 domain is

the most frequently identified in *vgrG* loci (Supplementary Table 1). We therefore explored its function in T6SS. Three *vgrG* loci encoding the DUF2345 domain were found in *Escherichia coli* PAR, *Pseudomonas aeruginosa* strain PAO1 and PS42 (Fig. 3a). Sequence comparison indicated that AKO63_2953 (VgrG$^{PAR}$), AKO63_2954 (DUF2345$^{PAR}$) and AKO63_2955 (M35$^{PAR}$), corresponding to the VgrG domain, the DUF2345 domain and the M35 (metallopeptidase) toxin domain of PA0262 (VgrG2b$^{PA}$), respectively. Similarly, Q094_05019 (VgrG$^{PS}$) of *P. aeruginosa* PS42 encodes VgrG domain, whereas Q094_05020 encodes N-terminal DUF2345 domain and C-terminal M35 domain. AlphaFold v2.0 predicted that VgrG$^{PAR}$, VgrG$^{PS}$ and VgrG domain of VgrG2b$^{PA}$ have the same conformation

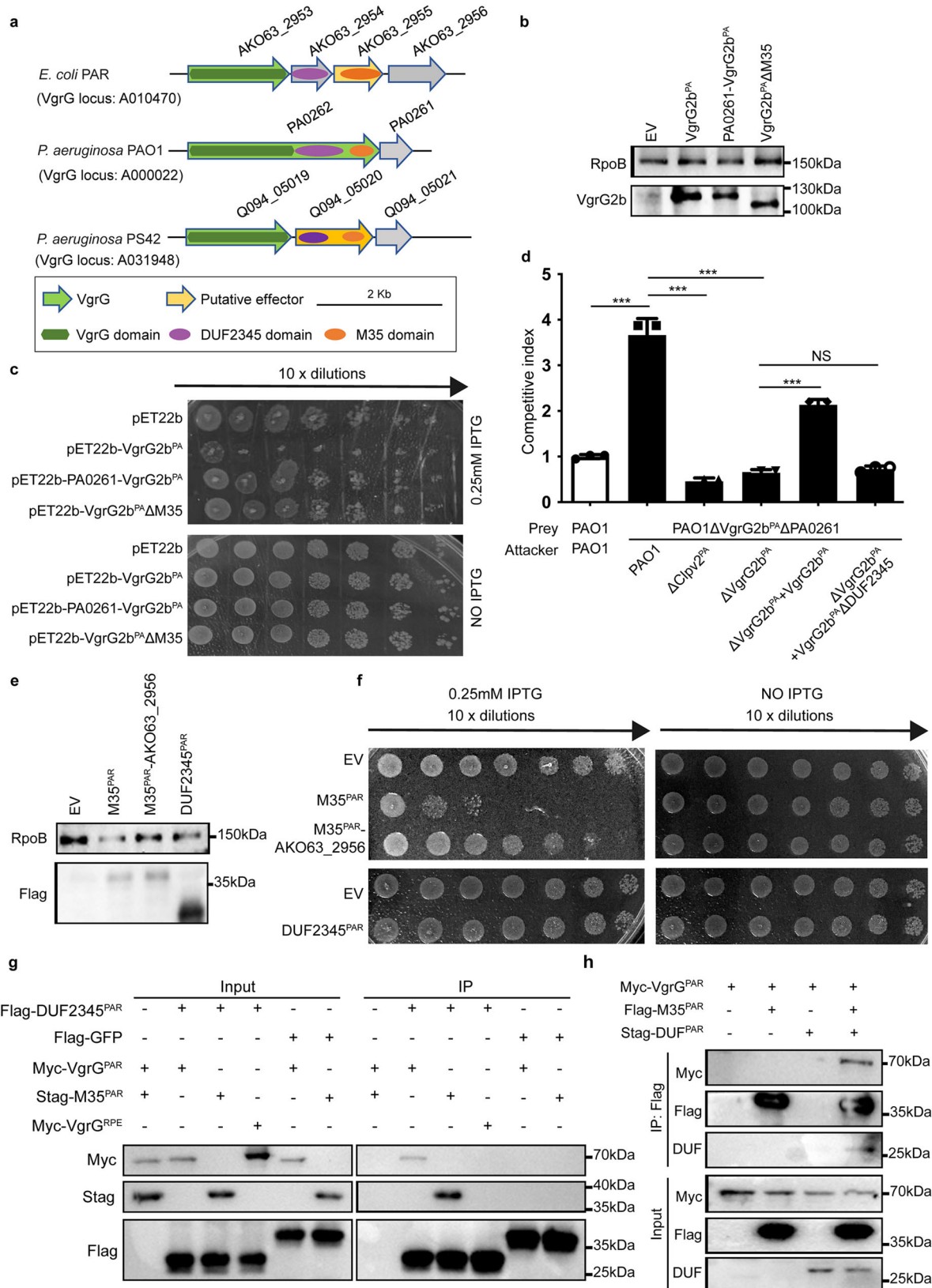

(Supplementary Fig. 5a). Further, *E.coli* locus (VgrG[PAR], DUF2345[PAR] and M35[PAR]), PS42 locus (VgrG[PS] and Q094_05020) and VgrG2b[PA] form similar trimmer structure, which implies that these three complexes might endow similar biological functions (Supplementary Fig. 5b). As these three loci encode VgrG, toxin and immunity proteins, we speculate that

DUF2345 maybe involved in the interaction between VgrG and its cognate effector.

Wood et al. showed that VgrG2b[PA]-PA0261 constitutes a T6SS antibacterial effector-immunity pair[30]. *E. coli* toxicity assay was used to test whether the DUF2345 domain in VgrG2b[PA] is toxic to bacteria (Fig. 3b, c). As expected, overexpressed in *E. coli*,

**Fig. 3 DUF2345 domain assist the delivery of T6SS effector. a** The *vgrG* loci of *E. coli* PAR, *P. aeruginosa* PAO1 and PS42. **b** *E. coli* expressing VgrG2b[PA] or its truncated mutant VgrG2b[PA]ΔM35 were detected by Western blot. Anti-RpoB is lysis control. **c** Survival of *E. coli* expressing VgrG2b[PA] or its truncated mutant VgrG2b[PA]ΔM35 in pET22b. Ten-fold serial dilutions of cultures were spotted on LB agar containing the stated concentrations of IPTG and grown for 24 h. The image is representative of three independent experiments. **d** Intraspecies *P. aeruginosa* competition assay between the ΔVgrG2b[PA]ΔPA0261 strain and various isogenic attacker strains at 37 °C for 24 h. Competition assay between the parental strain (PAO1) and itself (gray) is the internal control. The values and error bars represent the mean ± SD (*n* = 3 biological replicates). A one-way ANOVA with Dunnett's test was employed using the parent versus prey competition as the comparator (*$p < 0.05$; ns, not significant). **e** *E. coli* expressing M35[PAR], AKO63_2955-2956 or DUF2345[PAR] were detected by western blot. Anti-RpoB is lysis control. **f** Survival of *E. coli* expressing M35[PAR], AKO63_2955-2956 or DUF2345[PAR] in pET22b. Ten-fold serial dilutions of cultures were spotted on LB agar containing the given concentrations of IPTG and grown for 24 h. The image is representative of three independent experiments. **g** Interactions between DUF2345[PAR] and VgrG[PAR] or M35[PAR]. Shown here are immunoblots of lysates (total) and immunoprecipitates with anti-FLAG affinity beads (IP: FLAG) of DUF2345[PAR] transformed with empty vector or a plasmid encoding Myc-tagged VgrG[PAR] or S-tagged M35[PAR]. GFP and VgrG[PRE] are control proteins. **h** DUF2345[PAR] mediates the interaction between VgrG[PAR] and M35[PAR]. Shown here are immunoblots of lysates (total) and immunoprecipitates with an anti-FLAG affinity beads (IP:FLAG) of M35[PAR] transformed with a plasmid encoding either Myc-tagged VgrG[PAR] or S-tagged DUF2345[PAR].

---

VgrG2b[PA] exhibited acute toxicity and co-expression of the immunity gene (*PA0261*) relieved this growth defect. Crucially, truncation of the M35 domain of VgrG2b[PA] restored growth, which indicated that DUF2345 in itself is not toxic to *E. coli*. Intraspecies *P. aeruginosa* competition assays were also performed to determine whether the DUF2345 domain could affect the function of VgrG2b[PA] (Fig. 3d). Although the ΔVgrG2b[PA]ΔPA0261 strain exhibited a significant growth disadvantage against the wildtype PAO1 strain, it could no longer be outcompeted by both ΔClpV2[PA] and ΔVgrG2b[PA] attacker strain. Notably, compared with the wildtype *vgrG2b[PA]* gene, the complement of *vgrG2b[PA]*ΔDUF2345 could not restore the growth advantage of the attacker strain. Further, although the secretion of Hcp (the T6SS inner stylet protein) was not affected, complemented in the ΔVgrG2b[PA] strain, VgrG2b[PA]ΔDUF2345 could only be detected in the cells, but not in the supernatant (Supplementary Fig. 6a). In addition, the production of VgrG2b[PA]ΔDUF2345 was still detrimental to *E. coli* when it remains in the periplasm (Supplementary Fig. 6b, c). Therefore, it is clear that the DUF2345 domain disturbs the antibacterial ability of VgrG2b[PA] by ablation of its secretion.

We subsequently explored the function of DUF2345 when encoded as a distinct gene, which is within the locus containing *vgrG[PAR]*, *M35[PAR]*, along with the cognate immunity protein (Fig. 3a). *E. coli* toxicity assay demonstrated that M35[PAR] exhibited bacterial killing activity, which was inhibited by its immunity protein (Fig. 3e, f). Consistent with the results of Fig. 3c, expression of DUF2345[PAR] in isolation had no deleterious effect on bacterial growth (Fig. 3f). Immunoprecipitation assays of proteins co-expressed in *E.coli* confirmed that DUF2345[PAR] can specifically bind VgrG[PAR] and M35[PAR], but not VgrG[RPE] (VgrG in *Burkholderia* sp. RPE67) (Fig. 3g). Importantly, M35[PAR] could not interact with VgrG[PAR] in the absence of DUF2345[PAR] (Fig. 3h). These results implied that DUF2345[PAR] is involved in the interaction between VgrG[PAR] and M35[PAR] to assist the loading of M35[PAR] on the T6SS spike.

Taken together, DUF2345 domain is indispensable for the delivery of its cognate toxin via fusion at the C-terminus of VgrG or encoded as a single gene.

**Linkage between diverse DUF2345 genes and downstream toxins**. Considering that DUF2345 is encoded as either a fusion at the C-terminus of VgrG or a distinct gene downstream of *vgrG*, we then investigated whether the sequences of VgrG domains showed a correlation with those of DUF2345. An iterative procedure was devised to hierarchically cluster the 52,277 VgrG domains and their cognate DUF2345 domains, respectively. At the 30% amino-acid sequence similarity cutoff, VgrG domains form three major clusters and ten outliers,

whereas DUF2345 domains were classified into 37 distinct groups (Supplementary Fig. 7). These findings imply that, compared to the relatively conserved VgrG domains, the sequences of DUF2345 domains exhibited higher diversity.

As we demonstrated above, DUF2345 is involed in the interaction between VgrG and the toxin protein. To further delve into this, we performed a Sankey analysis to investigate the relationship between DUF2345 domains and their downstream toxins in greater detail. It is interesting to note that most of DUF2345 clusters showed an obvious taxon-specific distribution and correlated well with their downstream toxins (Fig. 4). Meanwhile, we also noticed that there are some toxins which correlated to more than one of DUF2345 clusters, such as Lyz-like and DUF2235 domains. To test whether this is a result of the intrinsic sequence diversity of these toxins, an iterative procedure was applied to further subdivide these toxin groups. As expected, the sub-clusters of Lyz-like and DUF2235 domains also correlated well to DUF2345 groups (Supplementary Fig. 8). Thus, our data reveals that, DUF2345 domains exhibit high sequence diversity and correlate well with their downstream toxins.

**LysM homologs assist the interaction between VgrG and downstream effector**. Absent from T6SS, LysM containing protein is one of the core components of eCIS, which shares several key homologous proteins in common with T6SS and forms a similar architecture[31,32]. Therefore, it is fascinating that our systematic screening implied that LysM domain is likely to be functional in T6SS.

Figure 5a showed a *vgrG* loci encoding a LysM containing protein in *Ketobacter alkanivorans* GI5. *E. coli* toxicity assay showed that Kalk_10455 exhibited acute toxicity and co-expression of Kalk_10450 relieved this growth defect, which indicated that Kalk_10450 is an immunity protein against Kalk_10455 (Fig. 5b, c). Notably, Kalk_10465 (VgrG[G15]) and Kalk_10460 (LysM[G15]) exhibited no toxicity when they were expressed in *E. coli* (Fig. 5c). Although immunoprecipitation assays of proteins co-expressed in *E.coli* confirmed that Kalk_10455 specifically binds LysM[G15] and VgrG[G15], Kalk_10455 could not bind VgrG[G15] in the absence of LysM[G15] (Fig. 5d).

BRPE67_05220 in *Burkholderia* sp. RPE67, which includes both LysM[RPE] and NLPC_P60[RPE] domain, was used to further explore the function of LysM domain (Fig. 5a). *E. coli* toxicity assays demonstrated that BRPE_05220 exhibited bacterial killing activity. Moreover, expression of NLPC_P60[RPE] domain in isolation had deleterious effect on bacterial growth, which was inhibited by BRPE_05230 (Fig. 5e, f). Further, immunoprecipitated wildtype BRPE_05220, but not LysM truncated in

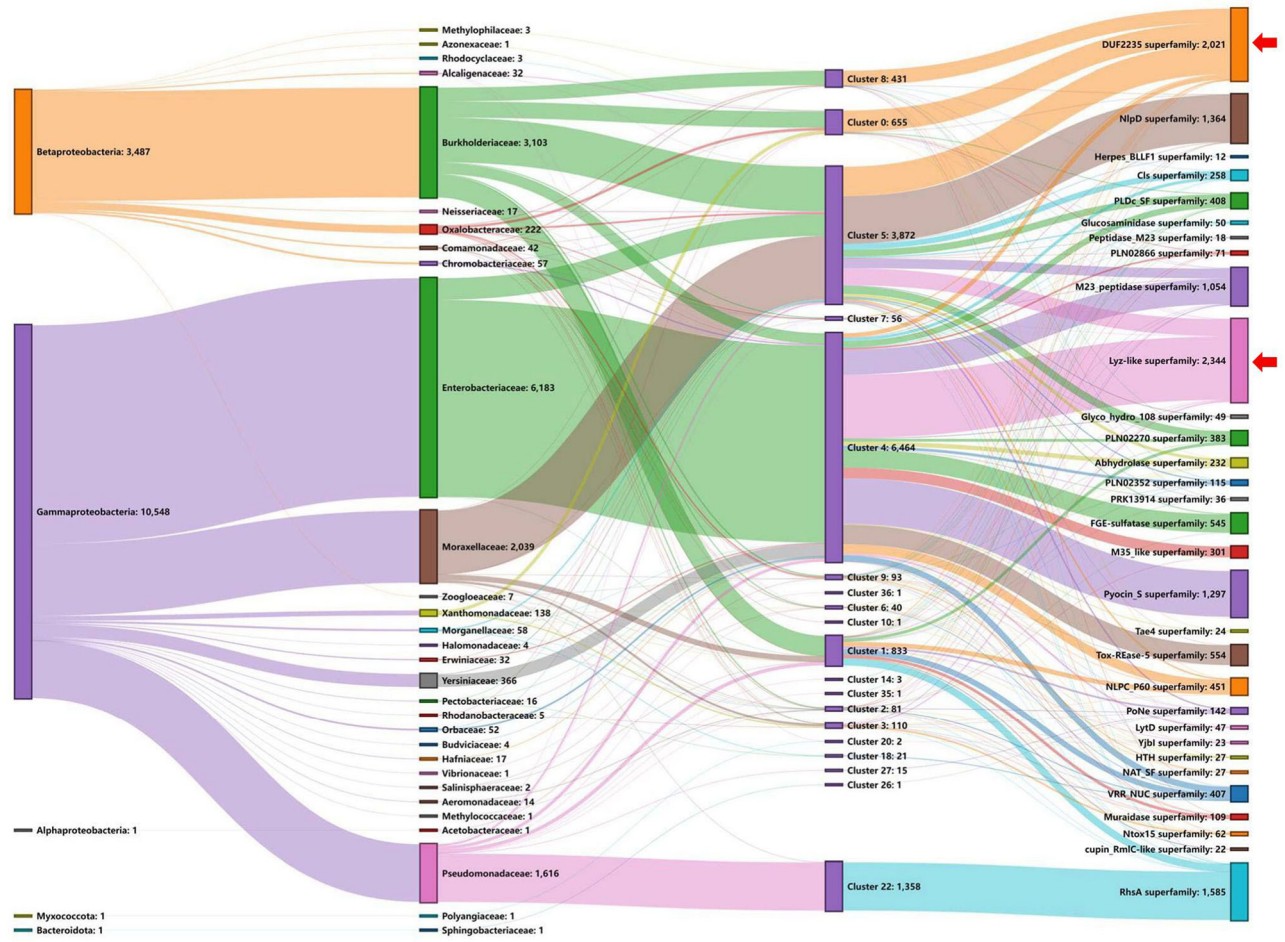

**Fig. 4 Correlation of DUF2345 domains and downstream toxins.** A Sankey diagram showing the relationship between bacterial phylum/class, family, the corresponding DUF2345 clusters and the downstream toxin domain families (from left to right). Only DUF2345-encoding loci with adjacent known toxin domains were included. Loci from genomes without necessary taxa information were excluded. The number of sequences involved in each node is given after the node name. The red arrows on the right indicate some toxins which were linked to more than one DUF2345 clusters.

BRPE_05220 (BRPE_05220ΔLysM), interacted with BRPE_05210 (VgrG $^{RPE}$) (Fig. 5g).

AlphaFold v2.0 predicted that BRPE_05210 (VgrG) and BRPE_05220 (LysM and NLPC_P60) form similar trimmer structure with VgrG2b$^{PA}$, which implied that LysM may mediate the interation between VgrG and toxin (Supplementary Fig. 9). Further, the LysM domain phylogenetic analysis revealed the diversity of T6SS-related LysM domains, which is evolutionarily distinct from the phage-/eCIS-associated LysM domains (Supplementary Fig. 10).

In sum, encoded at downstream of LysM containing gene or fused at the C-terminal of LysM domain, toxin interacts with VgrG in a LysM dependent manner implying LysM may assist the loading of its cognate effector onto the secretion apparatus.

**Six concerved domains are widely distributed in *vgrG* loci of various bacteria.** The DUF2345 containing proteins exhibit specific correlation with their downstream diverse toxins (Fig. 4). A similar Sankey analysis was performed to investigate the relationship between the other five identified conserved domain families along with the confirmed co-effector (MIX) and their downstream toxins (Supplementary Fig. 11). Notably, most of the characterized toxin domains showed an obvious domain specific distribution with limited exceptions. For instance, as polymorphic toxins, RHS-containing proteins encode variable C-terminal toxic domains with conserved N-terminal RHS domain[13]. Most of the

Rhs superfamily are linked to FIX-like (cl41761) and 5 (cl33691) domains. LysM domains are mainly correlated with Lyz_like, NlpD and NLPC_P60 superfamilies. As these domain families identified in this study, including FIX-like (cl41761), LysM (cl21525), 5 (cl33691), PG_binding (cl38043) and PHA00386 (cl30808), share a similar genetic organization and correlation with downstream toxins as the DUF2345 domain, it is reasonable to speculate that they would also function in the T6SS effector discrimination.

The overall distribution of the six conserved domain families was then analyzed (Fig. 6). It is interesting to note, these families were not evenly encoded among different bacterial families. For example, although DUF2345 domains are widely distributed among *Proteobacteria* genomes, they are rarely encoded in the genomes of *Vibrionaceae* and *Rhodospirillaceae* bacterial families. In contrast, the PG_binding_1 domain is limited to the genomes of γ-proteobacteria, including the families of *Chromatiaceae*, *Sinobacteraceae* and *Vibrionaceae*. In general, although these conserved domains are widely encoded among various bacteria, their distributions exhibit obvious taxonomic specificity, which is coincident with their corresponding cognate effectors as shown in Fig. 4 and Supplementary Fig. 11.

## Discussion
Widespread among the Gram-negative bacteria, T6SS employs diverse effectors to manipulate neighboring bacterial and/or

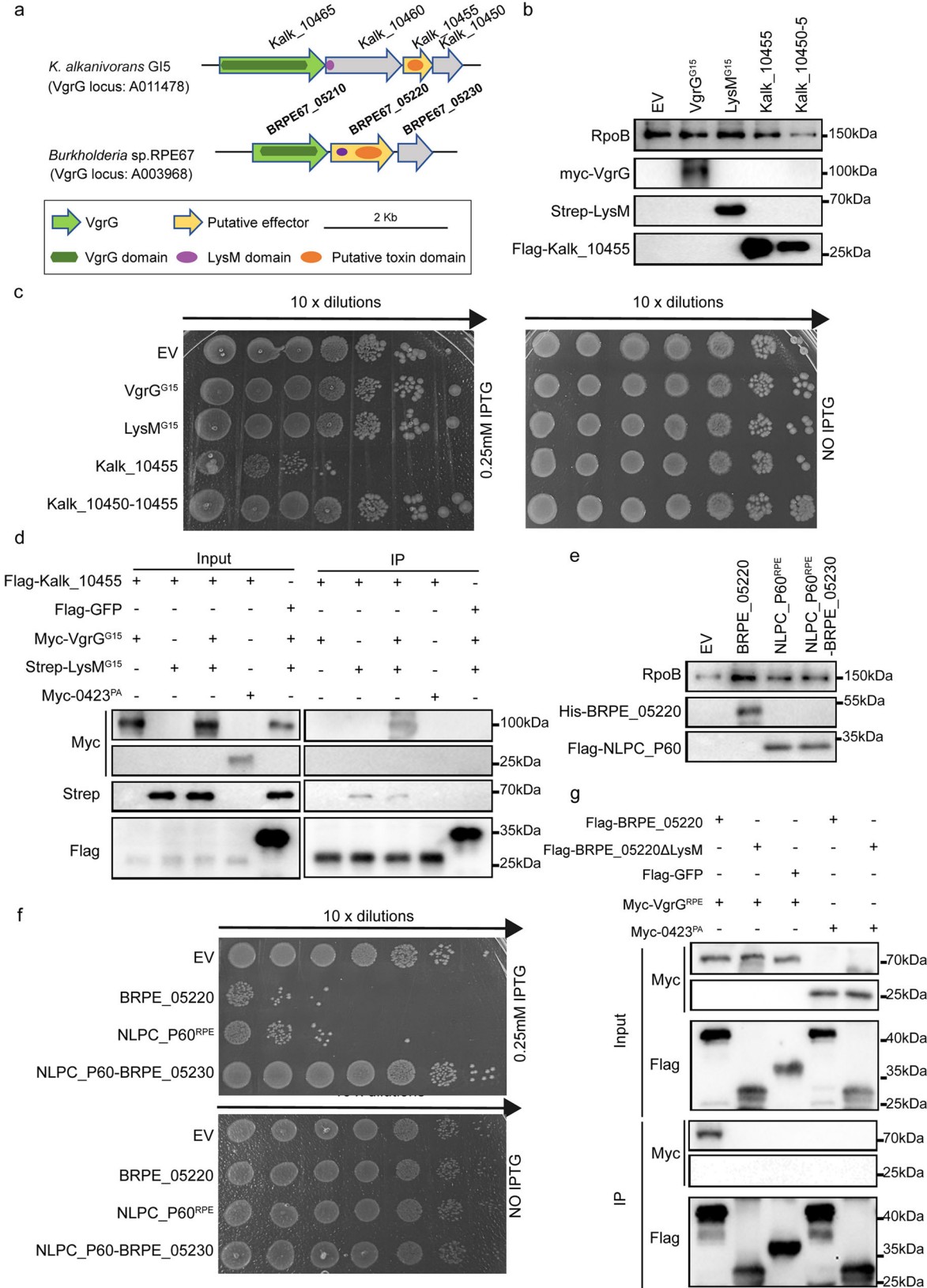

eukaryotic cells. As an essential component of T6SS injection apparatus, VgrG can deliver both specialized effectors and cargo effectors. T6SS adaptor (DUF4123, DUF2169 and DUF1795) and co-effector (MIX) genes have been found encoded adjacent to *vgrG*[16–19]. Encoded as a single protein or fused at the N-terminus of effector, a MIX containing protein is secreted with its cognate

effector for recipient bacterial killing, which is apparently different from classic adaptor proteins[20,21]. In this study, we created an open access VgrG database and identified six conserved domains, which exhibit similar genetic-context as MIX. Further bioinformatic and experimental approaches revealed the common characteristics of these six conserved domains.

**Fig. 5 LysM homologs assist the interaction between VgrG and the downstream effector. a** The *vgrG* loci of *Ketobacter alkanivorans* GI5 and *Burkholderia sp.* RPE67. **b** Immunoblots demonstrating the expression of VgrG2b[G15], LysM[G15] and Kalk_10455 in *E. coli*. Anti-RpoB is lysis control. **c** Survival of *E. coli* expressing VgrG[G15], LysM[G15] and Kalk_10455 in pETduet. Ten-fold serial dilutions of cultures were spotted on LB agar containing the stated concentrations of IPTG and grown for 24 h. The image is representative of three independent experiments. **d** Interactions between VgrG[G15], LysM[G15] and Kalk_10455. Shown here are immunoblots of lysates (total) and immunoprecipitates with anti-FLAG affinity beads (IP: FLAG) of Kalk_10455 and GFP transformed with a plasmid encoding Myc-tagged VgrG[G15] or Strep-tagged LysM[G15]. 0423[PA] is control protein. **e** Immunoblots demonstrating the expression of BRPE_05220 and NLPC_P60 domain in *E. coli*. Anti-RpoB is lysis control. **f** Survival of *E. coli* expressing BRPE_05220 and NLPC_P60[RPE] domain in pETduet. Ten-fold serial dilutions of cultures were spotted on LB agar containing the stated concentrations of IPTG and grown for 24 h. The image is representative of three independent experiments. **g** LysM domain mediates the interaction between VgrG[RPE] and BRPE_05220. Shown here are immunoblots of lysates (Input) and immunoprecipitates with an anti-FLAG affinity beads (IP:FLAG) of BRPE_05220 or BRPE_05220ΔLysM transformed with a plasmid encoding either Myc-tagged VgrG[RPE] or Myc-tagged 0423[PA]. 0423[PA] is control protein.

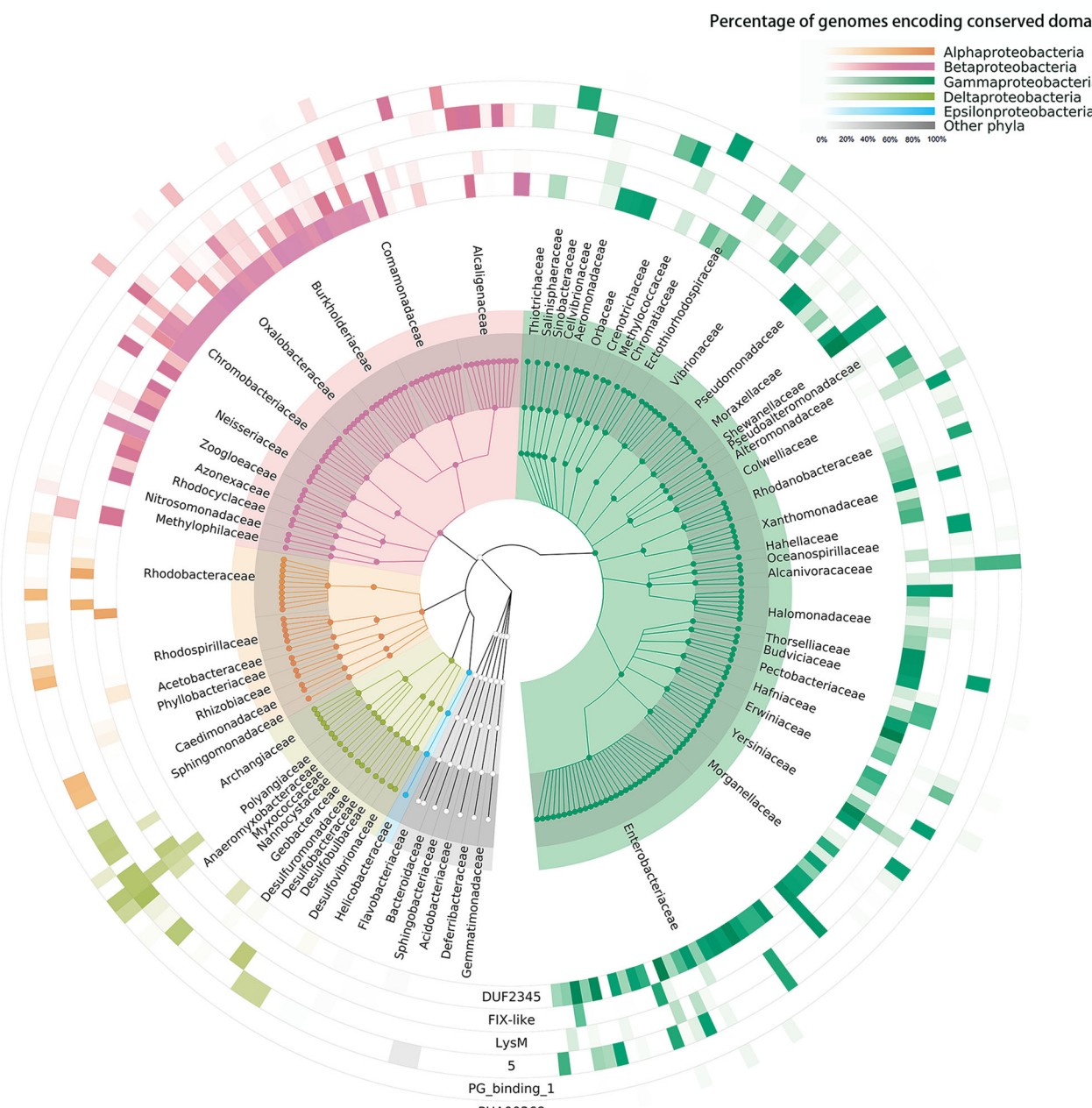

**Fig. 6 The taxonomic distribution of six conserved domains encoded within the *vgrG* loci.** Only taxa with genomes encoding at least one of the six conserved domains within the *vgrG* loci are shown for brevity. A total of 55,228 *vgrG* loci are included, but genomes without known assigned genus are excluded. The circles represent phylum, class, order, family and genus from inner to outer, and are color-coded by phylum/class (key). The family names are given outside the taxonomic tree. The outer heatmaps represent the percentage of genomes encoding the corresponding conserved domains for each genus (key).

To date, the structures of several VgrG proteins have been reported, which exhibits a canonical trimeric spike-forming region containing gp27 and gp5-like domains[6,33]. In addition to the conserved VgrG domain, the enteroaggregative *E. coli* (EAEC) VgrG (PDB 6sk0) contains a DUF2345 domain which adopts a β-helical fold similar to the gp5-like domain[6]. Previous studies reported that, fused at the C-terminus of VgrG, both DUF2345 and its downstream TTR domain are involved in the specific interaction with their effectors and DUF2345 proteins and so was termed as an internal adaptor of T6SS cargo effectors[6,29,30]. Our results revealed that DUF2345 domains correlated well with their downstream toxins, which further confirmed these previous studies. In the case of fusion at the C-terminus of VgrG, DUF2345 is inevitably to be secreted by T6SS. Whether the upstream and downstream regions of DUF2345, i.e. TTR, are always associated with DUF2345 would be interesting for further inverstigations.

LysM domains are found in a variety of enzymes involved in bacterial cell wall degradation and may have a general peptidoglycan binding function (CDD accession: cl21525). However, this domain is also encoded in fusion forms with phage baseplate components[34–36]. Interestingly, although LysM containing protein is an essential component of the eCIS particle, its protein homolog is not present in the structural components of T6SS, a subtype of contractile injection system[31,32]. Further phylogenetic analysis reveals the diversity of T6SS-related LysM domains that is distinct from the eCIS-related LysM domains. Considering LysM domains are not exclusively linked to T6SS-associated loci in bacterial genomes, it implies that some (if not all) of these domain-containing proteins might be multi-functional. Our study implied that, either when fused at the N-terminus of the effector or encoded as a distinct gene, LysM can assist loading the effector into the secretion apparatus through interaction with VgrG. Further structural and evolutionary studies are required to elucidate this new role of LysM in the bacterial secretion system.

Many bacterial secretion systems employ chaperones for the secretion of effectors[16,37]. T6SS co-effectors can be secreted in a T6SS dependent manner whereas adaptors are not translocated. RIX-containing proteins function as a secreted adaptor which enables the T6SS-mediated delivery of other cargo effectors by a previously undescribed mechanism[38]. Intriguingly, we found some *vgrG* loci encode both adaptor and conserved domains described in this study, such as MIX, FIX and LysM (Supplementary Figs. 2 and 3). Furthermore, more than one conserved domains are encoded in a single locus in some cases. For example, MIX and LysM domains are both encoded at the N-terminus of a *V. parahaemolyticus* T6SS effector (VPA1263)[21] and MIX, LysM and PG_binding domains are in one protein of *Vibrio* sp. CL6 (WP_061036948.1)[39]. Although our results indicated that LysM domain is involved in the interaction between VgrG and its cognate toxin, the phylogenetic tree showed that there are potential differences among VgrG, eCIS and phage-related LysM domains (Supplementary Fig. 10). Considering LysM domains are distributed among not only prokaryotes but also eukaryotes, its function in T6SS might be more complicated than expected, i.e., toxin in some cases. In addition, co-adaptor has been reported to improve the stability/expression of T6SS adaptor[40]. It is important to explore the detailed functions of these conserved domains especially when they are encoded in different configurations.

Our Sankey analysis indicated that these six conserved domains exhibit good correlation with their downstream toxins, likely reflecting the need for specific interactions with diverse toxins (Fig. 4 and Supplementary Fig. 11). Given the fact that these T6SS-associated conserved domains present in different forms in *vgrG* loci, a possible scenario could be the multiple origin and convergent evolution of these domains. Future investigations are required to further elucidate the function, diversity and evolution of these conserved domains in various bacterial genomes.

In this study, we used VgrG as a marker to identify conserved domains with multiple encoding forms. Of note, there are some T6SS-related loci encoding effectors without a partner *vgrG*[41]. It is formally possible that these effectors might employ conserved domains indentified in this study or adaptors/co-effectors encoded in other *vgrG* loci. Conversely, in such cases, it remains possible that potential conserved domains encoded within these loci were missed by the methods employed here. In addition, based on our screening criteria, we could only identify gene/domains adjacent to known toxins. We note that 34,272 of 52,277 (66%) DUF2345-encoding genes are in fact not linked to known toxin genes (Supplementary Table 1). Some potential domains would inevitably be omitted when there are no known toxin domain-encoding gene within the relevant *vgrG* loci.

Collectively, we identified six conserved domains with multiple encoding configurations, either as distinct genes or as fusion forms. These conserved domains may represent a common feature of T6SS effector recruitment, which is apparently different from other secretion systems.

## Materials and methods

**Bacterial strains and growth conditions**. Bacterial strains (Supplementary Table 2) were grown in Luria Bertani broth at 37 °C. *E. coli* DH5α (TransGen Biotech) was used for cloning procedures, and BL21(DE3) (TransGen Biotech) was used for protein expression and purification. *P. aeruginosa* strains were used for bacterial competition assays. All cultures were supplemented with antibiotics where necessary. The antibiotic concentrations were as followed, ampicillin (50 μg/mL), kanamycin (50 μg/mL), chloramphenicol (34 μg/mL), gentamicin (50 μg/mL) or tetracycline (50 μg/mL). Gene expression was induced with 1 mM IPTG.

**Identification of VgrG proteins encoded in bacterial genomes**. A total of 946 verified VgrG proteins (i.e., component TssI of T6SS) were downloaded from the SecReT6 database (accessed at July, 2020)[27]. The length of these VgrGs were surprisingly variable, between 69~1981 amino acids (aa). Therefore, based on a statistical histogram of their length distribution only 872 (92%) VgrGs with length of 601~1100 aa were retained and included as a positive control dataset of "typical" VgrG proteins (Fig. 1a). This dataset was then sent to the CD-search service of the Conserved Domain Database (CDD)[42] to search for conserved protein domains using the E-value cutoff $10^{-5}$ (accessed in July, 2020). Several VgrG-associated protein domains were successfully identified within one or more VgrGs, but the conserved domain recognized by CDD's COG3501 PSSM was the only conserved domain detectable in all of the 872 VgrG proteins. With this in mind, we used the presence of the VgrG domain (accession: COG3501) as a key marker for VgrG proteins in the follow-up study. However, previous studies revealed that the Afp8 proteins of eCISs elements are homologous to VgrG proteins. Indeed, we also identified 472 VgrG domain-encoding Afp8 proteins available from dbeCIS database[26], which were collected to serve as the negative control dataset for further VgrG screening.

In order to suppress potential false positives in future VgrG identification, we carefully analyzed the characteristics of the recognized VgrG domains within both of the aforementioned positive and negative control sets. Interestingly, 861 (99%) of the positive dataset encode a VgrG domain span of 451~750 aa, whereas only 10 (2%) of the negative dataset had VgrG domains that fit into this range (Fig. 1a). Therefore, we proposed an additional criterion of VgrG domain span in 451~750 aa to

maximally exclude the miss-identified Afp8 proteins while retaining the majority of true T6SS VgrGs. Subsequently, hidden Markov model (HMM) of COG3501 was used as the seed for a modified protein profile-based pipeline to systematically scan for VgrG proteins from a set of 133,722 publicly available bacterial genomes as previously proposed[43]. Of note, annotated Afp8 proteins available from the dbeCIS database were further excluded (Fig. 1a).

**Phylogenetic analysis of VgrG proteins**. Based on the aforementioned VgrG screening criteria each of the identified VgrG protein carries a 451~750 aa VgrG domain, which provided an ideal molecular marker for phylogenetic analysis. An in-house Perl script was utilized to extract the corresponding domain sequence from each VgrG. For brevity only one representative domain sequence was kept for VgrGs from the same bacterial genus with identical sequence. The remaining non-redundant domain sequences were fed to the Clustal-Omega program to generate multiple sequence alignment[44]. FastTree program v2.1 was then employed to construct the maximum-likelihood (ML) phylogenetic tree under the Whelan Goldman (WAG) models with gamma optimization[45]. The resulting phylogenetic tree was annotated and represented via the iTOL online server[46].

The current SecReT6 database has classified T6SS systems into eight types/subtypes based on the phylogeny of TssB (i.e., VipA/IglA)[27]. Therefore, we retrieved the existing classification information of the associated T6SS from the SecReT6 database for the 872 verified VgrGs. Then, we mapped the established type/subtype information onto the VgrG-based ML tree to propagate the current T6SS classification to all of the newly identified VgrGs.

**Construction of the VgrG database**. In order to facilitate future studies on the VgrG proteins as well as VgrG-related adaptors or effectors we constructed a publicly accessible online database, named dbVgrG (http://www.mgc.ac.cn/dbVgrG/) (Supplementary Table 3). The database integrates all results produced by the current study and illustrates the genetic organization and conserved domains of each VgrG locus. The database inherited most of the background MySQL data schema and foreground Perl CGI scripts developed in our previous studies after necessary customization[26,43]. To provide users necessary genomic context information of the identified VgrG proteins, we further included the coding genes within the upstream 10 kb and downstream 10 kb region (if they existed) of each *vgrG* locus. Moreover, the hmmsearch program from the HMMER3 package[47] was applied to identify the other 12 known components of T6SS based on the conserved domains summarized by the SecReT6 database[27].

**Molecular cloning**. PCRs were performed using a Biometra thermocycler using Fast Pfu DNA polymerase (TransGen Biotech). pETduet and RSFduet were used to express genes (Supplementary Table 2). Recombinant plasmids were constructed by pEASY®-Uni Seamless Cloning and Assembly Kit (TransGen Biotech). The tags were added by overlap extension PCR. Genes from *K.alkanivorans* GI5 and *Burkholderia* sp.RPE67 were synthesized and constructed to pETduet and RSFduet vectors by Biomed Biotech. DNA sequences and Primers used in this study were synthesized by RuiBiotech and listed in Supplementary Table 4.

In-frame deletion mutants were created as described previously[48]. Two DNA regions upstream (~700 bp) and downstream (~700 bp) of the target region were amplified by PCR using the specific primer pairs, which were cloned into pK18mobSacB. The plasmids were transformed into *E. coli* S17-1 λpir. After mobilization into *P. aeruginosa* by conjugation with *E. coli* S17-1 λpir, conjugants were selected using Luria Bertani broth supplemented with 200 μg/mL gentamicin and 34 μg/mL chloramphenicol. Plating onto LB agar containing 20% (w/v) sucrose selected for strains that had undergone double-recombination, excising the plasmid. Only colonies showing both kanamycin resistance and gentamicin sensitivity were selected. All mutants were confirmed by PCR and sequencing.

**Western blot analysis**. For western blots, samples resolved by SDS-PAGE were transferred onto PVDF membranes (Millipore), blocked with 5% skimmed milk for 30 min at room temperature. After that, the blots were incubated with the indicated primary antibodies overnight at 4 °C. The membrane was washed three times with Tris Buffered Saline with Tween 20 (TBST), then were incubated with horseradish peroxidase-conjugated (HRP-conjugated) secondary antibodies for 1 hr at room temperature. After a further three washes in TBST, the antibody-bound proteins were detected using SuperSignal West Femto Chemiluminescent Substrate Kit (Thermo Fisher Scientific, Waltham, MA) following the manufacturer's protocol.

**Secretion assays**. *P. aeruginosa* secretion assays were conducted as previously described with corresponding modifications[30,49]. Cultures were inoculated into 5 mL Luria Bertani broth at OD600 = 3.0. Supernatants were discarded from cells by centrifugation at $5000 \times g$ at 4 °C. Bacterial pellets were resuspended in sterile PBS and sub-cultured with 50-fold dilution in M9 medium to an $OD_{600}$ of 1.0. The supernatant and cells were separated by centrifugation. The supernatant was filtered through a 0.22 μm filter (Millipore, Billerica, MA, USA). One-fifth of the volume of 100% trichloroacetic acid was added into the supernatants. The fractions were incubated 2 h at −20 °C before centrifugation at $15,000 \times g$ and 4 °C for 30 min. The resulting protein pellets were washed three times with ice-cold acetone and resuspended in SDS protein sample buffer and then incubated at 100 °C for 10 min. For the cell pellet samples, 1 mL of culture was centrifuged and the total cells were harvested. Both the supernatant and pellet were analyzed by western blot analysis.

**Bacterial competition assays**. Bacterial competition assays were performed as described previously[30]. Briefly, donor and recipient strains were grown overnight and washed with sterile PBS. The mixture was spotted onto a nitrocellulose membrane overlaid onto a 3% LB-LS agar plate (LB low salt: 10 g peptone and 5 g yeast extract per liter). If needed, 0.5 mM IPTG was included in the medium for induction of cloned genes. The plates were incubated at 37 °C. Bacterial cultures were scraped off and suspended in 1 mL LB medium. After serial dilution, donor and recipient colonies were counted on selective LB agar plates and changes of the donor-recipient ratios were determined.

**Bacterial intoxication assays**. Overnight cultures of *E. coli* strain BL21 (DE3) containing pET22 b(+) expressing targeted proteins to the periplasm were serially diluted in LB medium at 10-fold. A total of 2.5 μL of this bacterial dilution was spotted on LB agar containing 0.25 mM IPTG. Images were taken after 24 h growth. *E. coli* strains harboring pET22 b(+) derivatives encoding both effector and immunity proteins were performed in a similar way.

**Immunoprecipitation**. PETduet and RSFduet plasmids were used for expression of VgrG, conserved domain-encoding proteins and effector proteins with FLAG/MYC/Strep tags in BL21 (DE3). After initial growth to 0.6 $OD_{600}$, 1 mM IPTG was added for a 16 h induction at 20 °C. Bacterial pellets were collected and

resuspended in IP-lysis buffer (50 mM Tris, pH 7.4, 250 mM NaCl and 6 mM KCl) containing a protease inhibitor PMSF. Cells were sonicated and centrifuged at 15,000 rpm for 20 min at 4 °C to remove cell debris. The supernatants were incubated with Anti-FLAG beads (Invitrogen). Bound proteins were eluted with elution buffer. Protein samples were separated by SDS-PAGE and analyzed by western blot. Anti-FLAG (MA1-91878; Invitrogen), anti-MYC (#2272; Cell Signaling Technology) and anti-Stag (#8476; Cell Signaling Technology) monoclonal antibody at a radio 1:1000 was used for Western blot analyses.

**VgrG downstream gene collection, domain screening and analysis**. Previous studies showed that VgrG-associated effectors, adaptors and co-effectors were usually encoded downstream of the *vgrG* gene in the adjacent genomic location. Therefore, we introduced the following operational criteria for the enrollment of genes downstream of *vgrG*: (i) for each *vgrG* locus a maximum of three continuous downstream genes were included for efficiency; (ii) valid downstream genes needed to be encoded on the same strand as *vgrG*; (iii) the intergenic region of all adjacent genes should <1 kb; and (iv) known components of the T6SS operon and any annotated pseudogenes were excluded. The CD-search of the CDD database[42] was then employed to screen for conserved domains among the 280,581 downstream genes in addition to the C-termini of VgrG proteins (i.e., sequences after each recognized VgrG domain if they existed) with the *E* value cutoff of $10^{-5}$. Additional domains were identified by the HHpred server manually[50]. The hierarchical sequence clustering of the VgrG domains, DUF2345 domains and other toxin domains were individually conducted by the CD-HIT v4.6.5[51] following the protocol described previously[43]. The relationships between domains/clusters were visualized by the SankeyMATIC web server (https://sankeymatic.com/).

**Scan of conserved domains within *vgrG* loci**. Many VgrG-related adaptors/co-effectors were expected to be flanked by the upstream *vgrG* and downstream toxin/effector genes. A collection of 928 experimentally verified exotoxins and effectors available from the well-established bacterial virulence factor database VFDB[52] were used to identify possible toxin/effector-related conserved domain families by CD-search (Fig. 2). Based on the genomic features of MIX domains, we developed an empirical criterion to screen three cases of possible conserved domains among the aforementioned downstream gene dataset. Case I: domains encoded by the VgrG (i.e., the domain is encoded between the VgrG domain and a toxin domain within the VgrG protein, or alternatively encoded by the C-terminus of VgrG where there is a downstream gene encoding a toxin domain in the same locus); Case II: domains encoded by a single distinct gene (i.e., the only identified domain encoded by a gene downstream of the *vgrG* that is then followed by a gene encoding toxin domain); and Case III: domains encoded by a toxin gene downstream of the *vgrG* (i.e., the domain is encoded as a fusion at the N-terminus of a downstream toxin gene, which has a toxin domain at its C-terminus).

After careful manual curation of the output domain lists for all three cases based on literature review and current annotations of reported domains, we finally accepted potential conserved domains that were present in not only a single gene but also either C-terminus of *vgrG* or N-terminus of toxin. In addition, to exclude possible false positives due to unexpected domain recognition we considered only domains with more than five instances present in at least two different bacteria genera. The taxonomic distribution of co-effector or conserved domains was visualized by the GraPhlAn tool[53].

**Statistical information**. Values of *n*, and the mean ± SD are reported and described in the figures and figure legends. GraphPad Prism 6 was used for all statistical analysis. All experiments reporting statistical significance were analyzed by one-way ANOVA with Dunnett's test. Statistical significance is denoted in figures by asterisks (*$p < 0.05$; **$p < 0.01$; ***$p < 0.001$).

**Reporting summary**. Further information on research design is available in the Nature Portfolio Reporting Summary linked to this article.

## Data availability
We used published data from NCBI to construct a publicly accessible online database, named dbVgrG (http://www.mgc.ac.cn/dbVgrG/). Other data supporting the findings of this study are included in the article and its Supplementary Information files, or from the corresponding authors upon request. Source data is provided in the Supplementary Data 1. The details of the 133,722 bacterial genomes screened in the study is provided in the Supplementary Data 2.

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

## Acknowledgements

This work was supported by the National Natural Science Foundation of China (31741008 to G.Y., 31970635 to J.Y. and 82103071 to C.W.); Beijing Hospitals Authority Youth Program (QML20230105 to C.W.); Research Foundation of Beijing Friendship Hospital, Capital Medical University (YYZZ202029) to C.W.; the CAMS Innovation Fund for Medical Sciences (2021-I2M-1-030) to Q.L.

## Author contributions

C.W., J.Y., and G.Y. conceived the project. C.W., M.J. and Y.S performed the experiments. M.C., L.C., and J.Y. constructed the database. C.W., M.C., Q.L., Y.W., L.C., S.C., N.R.W., J.Y., and G.Y. analyzed the data. C.W., M.C. N.R.W., J.Y., and G.Y. wrote the manuscript.

## Competing interests

The authors declare no competing interests.
