## [Peer Review File · Communications Biology]

Reviewers' comments:

Reviewer #1 (Remarks to the Author):

Disclaimer: I have seen this manuscript in an earlier version. Most of my comments and concerns have not been addressed, and so this review contains comments previously seen by the authors, and a few new ones.

In this manuscript by Wang et al, the authors perform a computational analysis to identify VgrG proteins in publicly-available genomes and use them to identify so-called "co-effectors", following notions proposed previously for a MIX domain-containing protein. The authors suggest that 6 domains conform to a co-effector-like pattern, and propose that these domains may be used as markers to identify additional T6SS effectors. Although the idea is intriguing and the generality of a secretion mechanism that includes a co-effector is worth studying, I have major concerns. These mainly relate to the premise of what should be regarded as "co-effector" and the interpretation of the computational findings, as well as to the lack of experimental support for protein secretion.

Major concerns:

- 1) Term-wise, the previous report of a MIX domain-containing "co-effector" did not imply that all MIX domains should be considered as such, only that stand-alone MIX-containing proteins that enable the secretion of a toxin encoded by a different gene be. Therefore, the authors use the term "co-effector" in this work very promiscuously, and even inappropriately. This also means that the premise of using MIX domains as a pattern to identify "co-effectors" is incorrect.
- 2) Other than MIX and FIX-like, the other identified domains are not specific to T6SS-associated proteins. Therefore, their use as "markers" for T6SS effectors is questionable. They can only be used as such in the context of a T6SS auxiliary operon, but then the presence of a secreted T6SS component (VgrG, Hcp, or PAAR) already alludes to the presence of a downstream effector within the operon.
- 3) No experimental evidence is provided to support the claims of the domains' role as so-called "co-effectors" in effector secretion (co-IP is provided for LysM in *E. coli*, but no evidence for its effect on secretion in an actual T6SS model). DUF2345 is further investigated, yet its role in T6SS structure and effector secretion is already known and published. MIX and FIX-like domains have been reported and studied previously, and the only difference here is that they are named differently than in previous publications as "co-effectors".
- 4) Lines 337-339: If LysM domains (predicted to bind peptidoglycan) are found together with MIX domains in the same protein, I don't understand how the authors envision a scenario of two "co-effectors" working in one protein. Doesn't the presence of these two domains in the same protein indicate that they play different roles? Again, I have trouble with the functional definitions used here for different domains.
- 5) Lines 282-3: LysM correlates with toxin domains that target the bacterial periplasm. LysM is also predicted to be a peptidoglycan binding domain (line 324). This raises doubt regarding the assigned role for LysM as a "co-effector". Did the author try to predict the structure of a LysM-containing effector and found that LysM is predicted to mediate the interaction with VgrG? Is it possible that the protein truncations used in the co-IP experiments were simply not properly folded, and that is why a LysM domain truncation did not bind VgrG?

6) Regarding domain "5": Is it not considered an extension the VgrG? To my knowledge, this is mainly a structural extension of the VgrG beta-helix (similar to DUF2345, actually). This can be visualized by inputting a trimer of this domain into an alphafold multimer prediction.

7) Lines 108-111/Fig. 1B: There are two branches in the tree that are unmarked as belonging to a specific T6SS subtype. What are they? Their presence an unknown association makes it difficult to determine that "ML tree exhibited a similar overall topology regarding types/subtypes of T6SS operons as previously described".

Minor comments:

1) Line 62: Consider replacing "complicated" with "complex".

2) Line 63: the "tube-spike" is propelled out, not only the spike.

3) Fig. S1: It seems that the position of the gamma and delta proteobacteria is switched in the legend.

4) Lines 136-140: The rationale behind the conclusion that horizontal gene transfer supports a co-effector role it unclear.

Reviewer #3 (Remarks to the Author):

The authors present a concise search for novel co-effector domains of the type VI secretion system (T6SS), which was based on an in-silico screening for novel candidate domains, based on a set of rules, followed by experimental validation using protein expression and interaction assays. It is notable that the authors are successful in finding six novel domains families and managed to demonstrate that members of two of these families are required for T6SS delivery.

Overall, the manuscript presents a convincing demonstration that the VgrG-associated domains analyzed are indeed co-effectors, at least when fused to the toxin domains. The in-silico selection pipeline proves its efficiency with the experimental validation and is based on a sound procedure, but notably it depends on the information from the reference databases, VFDB and dbCIS, being correct. Although this could potentially be a problem, the selected targets suggest no evidence of problems in the reference database.

It should be noted that this work focus on a narrow target, concentrating on proving the existence of additional co-effector and/or adaptor domains. Such a result is indeed useful for the community and I would recommend this manuscript for publication after the author's review it to correct several minor issues in its presentation.

On the issue of originality, the authors present six new families but only demonstrate the adaptor function for two of these families, the first of which was already understood as playing the role of an adaptor by several authors. The LysM domain, to my knowledge, was never associated with protein-protein interactions between effectors and toxins and such a conclusion is, therefore, novel.

Notably, the lack of structural data showing the details of LysM-effector interactions is regrettable, as it would be a great addition to the literature and greatly strengthen the article. Experimental data on the other four adaptor domains, namely PG_binding_1, FIX-like, PHA00368 and 5), is also not presented, therefore hindering the full evaluation of the discovery pipeline.

Still, I think this results can wait and it is better to allow the larger community of readers to get to know the results presented in this manuscript, so that they can be evaluated.

The following minor issues should be addressed before publication:

1) In the abstract, at row 33, replace "and it is a distinct features of T6SS" by "and such co-effector domains are distinct features of T6SS"

2) Row 53, replace "Usually encoded adjacent" by "These domains are encoded adjacent"

3) Considering all figures, tables and text, including the supplementary files: update all clade names to match the most recent bacterial taxonomy standards, e.g. replace Deltaproteobacteria with Myxococcota.

4) About the RhsA family, referred at row 254 and in the figures, what defines this family? If you are considering all proteins containing RHS repeats, it should be noted that these proteins contain a large and diverse set of toxic domains at their C-terminal end, as shown in previous work about polymorphic toxins. It is therefore not a good idea to place all RHS-containing proteins in the same family, but instead recognize different families for each toxic C-terminal domain family. See Zhang, D., de Souza, R.F., Anantharaman, V. et al. Polymorphic toxin systems: Comprehensive characterization of trafficking modes, processing, mechanisms of action, immunity and ecology using comparative genomics. *Biol Direct* 7, 18 (2012). <https://doi.org/10.1186/1745-6150-7-18>.

5) With regard to Figure S5, I personally dislike pizza graphs, specially those with few categories, which tend to be just as informative as a small table. I suggest you replace this graph with another Sankey diagram that would display both the counts shown in figure S5 and the association of each vgrG loci type with subtypes of T6SS.

6) On rows 357-358, to emphasize the difference between a protein domain and the mathematical model that is used to represent and detect the domain, replace "but domain COG3501 of VgrG superfamily was the only conserved domain that were detectable in all of the 872 VgrG proteins." with "but the conserved domain recognized by CDD's COG3501 PSSM was the only conserved domain detectable in all of the 872 VgrG proteins."

7) On row 370, replace "remaining" with "retaining".

8) Include all alignments used for phylogeny in the supplement, together with the final tree files in both Newick and Nexus formats.

9) For the data presented in figures 3 and 5, all western blot photos should be provided as raw photos in the supplement. This is required so one can see all data in a single photo with the molecular weight references. It is essentially not possible to criticize these results without the raw data.

10) The authors should replace all gene and protein names and , locus tags used to identify the experimentally verified proteins with the locus tag present in dbVgrG. This is essential since the names used in the blotting and other analysis cannot be confidently identified without using dbVgrG.

11) The relationship between the primers (Table S2) and their targets is not clear to me: I could not find even a subsequence of the primers, or their reverse complement, matching the genomes and genes they were supposed to amplify. Also, for the NlpC/P60 gene from Burkholderia sp. RPE67, the reverse primers sequence is missing.

Reviewer #1 (Remarks to the Author):

Disclaimer: I have seen this manuscript in an earlier version. Most of my comments and concerns have not been addressed, and so this review contains comments previously seen by the authors, and a few new ones.

In this manuscript by Wang et al, the authors perform a computational analysis to identify VgrG proteins in publicly-available genomes and use them to identify so-called “co-effectors”, following notions proposed previously for a MIX domain-containing protein. The authors suggest that 6 domains conform to a co-effector-like pattern, and propose that these domains may be used as markers to identify additional T6SS effectors. Although the idea is intriguing and the generality of a secretion mechanism that includes a co-effector is worth studying, I have major concerns. These mainly relate to the premise of what should be regarded as “co-effector” and the interpretation of the computational findings, as well as to the lack of experimental support for protein secretion.

We appreciate the valuable comments raised by the Reviewer. The co-effector (MIX) assists in the interaction between VgrG and the cognate effector and is secreted along with the effector via the T6SS (Dar. *et al*, 2022, *EMBO Rep*). Recently, it was demonstrated that MIX domain encoded at the N-terminus of T6SS toxin could be secreted via T6SS and is necessary for the secretion of this toxin (Fridman. *et al*, 2022, *Microbiol Spectr*). Admittedly, the current knowledge of the MIX domain-containing “co-effector” is still limited. Although the idea of T6SS co-effector is intriguing and worth studying, we do agree with the Reviewer that it is inadequate to use MIX domain as a pattern to identify further “co-effectors” and the conclusion was overstated based on our current results.

Furthermore, the conserved domains identified here are supposed to be secreted when they are fused with the VgrG/toxin. Although the secretion and intraspecific competition of DUF2345 domain in *P. aeruginosa* PAO1 have been validated here, we agree with the Reviewer that similar experiments are also important for the validation of LysM domain in an actual T6SS model. We contacted several strain centers and groups. But unfortunately, we were not able to obtain the strains to perform these experiments. Nonetheless, as the

Reviewer suggested, we have added the alphafold predicted structure of a VgrG-LysM-toxin complex (Figure S9) and a LysM domain phylogenetic tree (Figure S10) in the revised manuscript to further explore the function of LysM.

To make the conclusions more convincing, we have modified our interpretation of the computational and experimental findings as conserved domains for the effector delivery of bacterial type VI secretion system in the revised manuscript.

According to the Reviewer's concerns, further modifications were made as following.

Major concerns:

1) Term-wise, the previous report of a MIX domain-containing “co-effector” did not imply that all MIX domains should be considered as such, only that stand-alone MIX-containing proteins that enable the secretion of a toxin encoded by a different gene be. Therefore, the authors use the term “co-effector” in this work very promiscuously, and even inappropriately. This also means that the premise of using MIX domains as a pattern to identify “co-effectors” is incorrect.

It is indeed an important point of our work. As we mentioned above, we agree with the Reviewer that it is not appropriate to term the conserved domains identified in this study as “co-effectors”. Both stand-alone and fused MIX domains are important for the secretion of the toxin (Dar. *et al*, 2022, *EMBO Rep*; Fridman. *et al*, 2022, *Microbiol Spectr*). The six domains identified here exhibit a similar genetic pattern as the MIX domain. Further, more experimental results confirmed that both DUF2345 and LysM domains are indeed involved in the interaction between VgrG and the cognate toxin. Thus, we have toned down the conclusion to make it more convincing.

2) Other than MIX and FIX-like, the other identified domains are not specific to T6SS-associated proteins. Therefore, their use as “markers” for T6SS effectors is questionable. They can only be used as such in the context of a T6SS auxiliary operon, but then the presence of a secreted T6SS component (VgrG, Hcp, or PAAR) already alludes to the

presence of a downstream effector within the operon.

We do agree with the Reviewer that these conserved domains are inappropriate to be applied as “markers” for T6SS effectors. Related descriptions in this manuscript have therefore been deleted.

3) No experimental evidence is provided to support the claims of the domains’ role as so-called “co-effectors” in effector secretion (co-IP is provided for LysM in *E. coli*, but no evidence for its effect on secretion in an actual T6SS model). DUF2345 is further investigated, yet its role in T6SS structure and effector secretion is already known and published. MIX and FIX-like domains have been reported and studied previously, and the only difference here is that they are named differently than in previous publications as “co-effectors”.

We very much agree with the Reviewer that secretion experiments in an actual T6SS model are important to further verify the function of LysM. However, as we explained above, we could not obtain the strains to perform these experiments. As the Reviewer may notice, with the exception of the genes from *P. aeruginosa* PAO1, all the rest of these genes in this study were synthesized for co-IP and *E. coli* toxicity assay.

In the revised manuscript, we also performed the alphafold prediction structure of VgrG-LysM-toxin complex and LysM domain phylogenetic tree to further explore the function of LysM. Thus, in accordance with the Reviewer’s comments and our results, we have restated the conclusion and revised the title as “Genome wide analysis revealed conserved domains for effector delivery of bacterial type VI secretion system”.

Although FIX-like domain has been previously reported as Marker of T6SS substrates, its function in T6SS is still not clear (Jana. *et al* 2019, *Nat commu*). Here, the bioinformatics analysis we proposed suggests it might play a similar role to the MIX domain.

Furthermore, although the role of the DUF2345 in T6SS is known, our data revealed that these conserved domains, including DUF2345, are encoded as both single and fusion

forms in T6SS. These genetic configurations may represent the generic feature of T6SS effector discrimination mechanism. We hope our results would further facilitate future investigations into the detailed functions of these conserved domains.

4) Lines 337-339: If LysM domains (predicted to bind peptidoglycan) are found together with MIX domains in the same protein, I don't understand how the authors envision a scenario of two "co-effectors" working in one protein. Doesn't the presence of these two domains in the same protein indicate that they play different roles? Again, I have trouble with the functional definitions used here for different domains.

Yes, we had mentioned this protein in the Discussion (Line 304-305 in the reviewed version). Although it is not a common scenario, two conserved domains encoded in one T6SS effector is still interesting. As the Reviewer mentioned, these two domains in the same protein may play different roles in T6SS. Except this case, we also found some known T6SS adaptors are encoded with MIX, FIX or LysM domain within the same *vgrG* loci (Fig S2; S3). All these cases indicated that T6SS may employ some more sophisticated loading mechanism than expected. Further investigations will be performed to explicit these cases in details in the future. In the revised manuscript, we have restated the conclusion that these conserved domains may assist for the T6SS effector delivery, which should be well supported by our results.

5) Lines 282-3: LysM correlates with toxin domains that target the bacterial periplasm. LysM is also predicted to be a peptidoglycan binding domain (line 324). This raises doubt regarding the assigned role for LysM as a "co-effector". Did the author try to predict the structure of a LysM-containing effector and found that LysM is predicted to mediate the interaction with VgrG? Is it possible that the protein truncations used in the co-IP experiments were simply not properly folded, and that is why a LysM domain truncation did not bind VgrG?

This part is presented in the Discussion (Lines 319-321) in the revised version. CDD

prediction suggests the LysM domain is Peptidoglycan binding domain as previous studies showed that LysM domain is encoded in a variety of enzymes (<https://www.ncbi.nlm.nih.gov/Structure/cdd/cddsrv.cgi?uid=451289>). However, as we mentioned, LysM domains are also encoded by some phages and eCIS particles as an essential component, but not in T6SS (Lines 241-243; 321-324). In the revised manuscript, we have added the LysM domain phylogenetic tree analysis (revised Fig S10), which reveals the diversity of T6SS-related LysM domains, evolutionarily distinct from the phage-/eCIS-associated LysM domains. It is also coincident with our Sankey analysis that LysM domains exhibit specific correlation with their cognate toxins (Figure S11).

We also agreed with the Reviewer that truncated proteins may not be properly folded. As the Reviewer suggested, we further performed the alphafold multimer prediction structure of VgrG-LysM-toxin complex (revised Fig S9), which indicated that LysM could mediate the interaction with VgrG and the cognate toxin.

Admittedly, it is overstated to claim LysM as T6SS co-effector based on our current experiment results. Modifications were made in the revised manuscript.

Figure S9. Model of the complex of BRPE_05230 and BRPE_05220 proteins in *Burkholderia* sp. RPE67.

Structure model of the VgrG^{RPE}-LysM^{RPE}-NLPC_P60^{RPE} complex monomer (left) and the trimer complex (right).

Figure S10. Phylogenetic tree of LysM domain.

A maximum-likelihood tree of the 2,415 VgrG-downstream, 737 eCIS-related (available from dbEIS) and 19 phage-encoded LysMs based on the sequences of their LysM domain (CDD accession: c121525). Identical sequences within each group were removed to retain only one representative for brevity. The tree was constructed by FastTree under Whelan Goldman (WAG) models with gamma optimization. The eCIS-related and phage-encoded LysMs are highlighted by blue and red branches, respectively. The tree scale represents substitutions per site.

6) Regarding domain “5”: Is it not considered an extension the VgrG? To my knowledge, this is mainly a structural extension of the VgrG beta-helix (similar to DUF2345, actually). This can be visualized by inputting a trimer of this domain into an alphafold multimer prediction.

As the Reviewer suggested, an alphafold multimer prediction was performed (Fig i). It showed that 5 domain is an extension the VgrG which is similar to DUF2345. Although 5

and DUF2345 domains belong to different family, our results implied that they have similar roles in T6SS, such as toxin discrimination.

Figure i Model of EC042_1585 protein (including VgrG domain and 5 domain) in *E. coli* 042.

7) Lines 108-111/Fig. 1B: There are two branches in the tree that are unmarked as belonging to a specific T6SS subtype. What are they? Their presence an unknown association makes it difficult to determine that “ML tree exhibited a similar overall topology regarding types/subtypes of T6SS operons as previously described”.

We can understand the concern of the referee. However, please note that the previous study by Li et al. included only 825 known T6SSs, whereas our current study collected 130,825 VgrGs from 45,041 Gram-negative bacterial genomes to produce the phylogenetic tree. We believe it is to be expected to reveal several unknown clades in the current larger dataset. As for the topology comparison of the previous and current trees, the following are the Figure S4 (Li. *et al* 2015, *Environ Microbiol*) from Li et al. (up) and the current manuscript revised Figure 1B (down, with the same color-code as the upper panel). It's more straightforward than before that our ML tree indeed exhibits a similar overall topology regarding types/subtypes of T6SSs as the previous one (upper panel).

Minor comments:

1) Line 62: Consider replacing “complicated” with “complex”.

Corrected (line 73)

2) Line 63: the” tube-spike” is propelled out, not only the spike.

Corrected (line 74)

3) Fig. S1: It seems that the position of the gamma and delta proteobacteria is switched in the legend.

Thanks for the careful inspection. The legend of Figure S1 was corrected accordingly.

Figure S1. The taxonomic distribution of 7,208 MIX domains encoded within the *vgrG* loci.

Only taxa with genomes encoding MIX-encoding *vgrG* loci are shown for brevity. Genomes without known assigned genus are excluded. The circles represent phylum, class, order, family and genus from inner to outer, and are color-coded by phylum/class (key). The family names are given outside the taxonomic tree. The outer heatmap represents the percentage of genomes encoding the MIX domain for each genus (key).

4) Lines 136-140: The rationale behind the conclusion that horizontal gene transfer supports a co-effector role is unclear.

We showed some horizontal acquisition or genetic deletion events of these conserved domains without *vgrG* neighborhood. Indeed, this preliminary data is not related to this study. For clarity, we deleted Figure S4 and the description in the revised version (line 152-156).

Reviewer #3 (Remarks to the Author):

The authors present a concise search for novel co-effector domains of the type VI secretion system (T6SS), which was based on an in-silico screening for novel candidate domains, based on a set of rules, followed by experimental validation using protein expression and interaction assays. It is notable that the authors are successful in finding six novel domain families and managed to demonstrate that members of two of these families are required for T6SS delivery.

Overall, the manuscript presents a convincing demonstration that the VgrG-associated domains analyzed are indeed co-effectors, at least when fused to the toxin domains. The in-silico selection pipeline proves its efficiency with the experimental validation and is based on a sound procedure, but notably it depends on the information from the reference databases, VFDB and dbCIS, being correct. Although this could potentially be a problem, the selected targets suggest no evidence of problems in the reference database. It should be noted that this work focuses on a narrow target, concentrating on proving the existence of additional co-effector and/or adaptor domains. Such a result is indeed useful for the community and I would recommend this manuscript for publication after the author's review it to correct several minor issues in its presentation.

On the issue of originality, the authors present six new families but only demonstrate the adaptor function for two of these families, the first of which was already understood as playing the role of an adaptor by several authors. The LysM domain, to my knowledge, was never associated with protein-protein interactions between effectors and toxins and such a conclusion is, therefore, novel.

Notably, the lack of structural data showing the details of LysM-effector interactions is regrettable, as it would be a great addition to the literature and greatly strengthen the article. Experimental data on the other four adaptor domains, namely PG_binding_1, FIX-like, PHA00368 and 5), is also not presented, therefore hindering the full evaluation of the discovery pipeline.

Still, I think these results can wait and it is better to allow the larger community of readers to get to know the results presented in this manuscript, so that they can be evaluated.

We appreciate the Reviewer consider this manuscript “is indeed useful for the community” and would recommend it for publication after minor revision.

As the Reviewer 1 suggested, we have made our conclusion more convincing by modifying the interpretation of our findings, changing the title to “Genome wide analysis revealed conserved domains for effector delivery of bacterial type VI secretion system”. Modifications and proofreading are also made throughout the manuscript.

The following minor issues should be addressed before publication:

1) In the abstract, at row 33, replace "and it is a distinct features of T6SS" by "and such co-effector domains are distinct features of T6SS"

Corrected (line 39-40)

2) Row 53, replace "Usually encoded adjacent" by "These domains are encoded adjacent"

Corrected (line 60)

3) Considering all figures, tables and text, including the supplementary files: update all clade names to match the most recent bacterial taxonomy standards, e.g. replace Deltaproteobacteria with Myxococcota.

The related taxa in revised Fig4 and Fig S11 were updated based on the most recent information from the NCBI taxonomy database as suggested.

Figure 4. Correlation of DUF2345 domains and downstream toxins.

A Sankey diagram showing the relationship between bacterial phylum/class, family, the corresponding DUF2345 clusters and the downstream toxin domain families (from left to right). Only DUF2345-encoding loci with adjacent known toxin domains were included. Loci from genomes without necessary taxa information were excluded. The number of sequences involved in each node is given after the node name. The red arrows on the right indicate some toxins which were linked to more than one DUF2345 clusters.

Figure S11. Sankey diagram showing the relationship between bacterial phylum/class, family, the VgrG-related conserved domains and the downstream toxin domain families (from left to right).

For brevity DUF2345 was excluded as it was already analyzed in Figure 4. Loci from genomes without necessary taxa information were excluded. The number of sequences involved in each node is given after the node name.

4) About the RhsA family, referred at row 254 and in the figures, what defines this

family? If you are considering all proteins containing RHS repeats, it should be noted that these proteins contain a large and diverse set of toxic domains at their C-terminal end, as shown in previous work about polymorphic toxins. It is therefore not a good idea to place all RHS-containing proteins in the same family, but instead recognize different families for each toxic C-terminal domain family. See Zhang, D., de Souza, R.F., Anantharaman, V. et al. Polymorphic toxin systems: Comprehensive characterization of trafficking modes, processing, mechanisms of action, immunity and ecology using comparative genomics. *Biol Direct* 7, 18 (2012). <https://doi.org/10.1186/1745-6150-7-18>.

Thank you for your comments. The RhsA superfamily was defined by the CDD based on the COG3209 conserved domain, which contains 28 RHS repeats (<https://www.ncbi.nlm.nih.gov/Structure/cdd/cddsrv.cgi?uid=225750>).

RHS-containing proteins are polymorphic toxins with conserved N-terminal RHS domain and variable C-terminal toxic domains (Zhang, *et al*, 2012, *Biol Direct*), most of which are not defined as toxin domains by the CDD and VFDB. In this study, as we only analyzed known toxin protein/domains downstream of *vgrG* loci, RHS proteins are considered as one toxin family in Sankey analysis. To avoid confusion, we added “as polymorphic toxins, RHS-containing proteins encode variable C-terminal toxic domains with conserved N-terminal RHS domain.” (Line 276-277).

5) With regard to Figure S5, I personally dislike pizza graphs, specially those with few categories, which tend to be just as informative as a small table. I suggest you replace this graph with another Sankey diagram that would display both the counts shown in figure S5 and the association of each *vgrG* loci type with subtypes of T6SS.

Figure S5 (revised Figure S4) was updated to a Sankey diagram style as suggested.

Figure S4. A Sankey diagram showing the relationship between the adaptor/MIX/conserved domains encoding *vgrG* loci (left) and the known subtypes of T6SS (right).

Only the three reported adaptor domains (i.e. DcrB, DUF4123 and DUF2169), the MIX domain, and the six conserved domains identified in current study among all of the 130,825 loci analyzed. The *vgrG* loci calculation was done progressively in the order of adaptors, MIX domain, and conserved domains.

6) On rows 357-358, to emphasize the difference between a protein domain and the mathematical model that is used to represent and detect the domain, replace "but domain COG3501 of VgrG superfamily was the only conserved domain that were detectable in all of the 872 VgrG proteins." with "but the conserved domain recognized by CDD's COG3501 PSSM was the only conserved domain detectable in all of the 872 VgrG proteins."

Corrected (line 388-390)

7) On row 370, replace "remaining" with "retaining".

Corrected (line 403)

8) Include all alignments used for phylogeny in the supplement, together with the final tree

files in both Newick and Nexus formats.

The multiple sequence alignment and ML tree files in both Newick and Nexus formats are provided as supplementary file x as suggested.

9) For the data presented in figures 3 and 5, all western blot photos should be provided as raw photos in the supplement. This is required so one can see all data in a single photo with the molecular weight references. It is essentially not possible to criticize these results without the raw data.

All raw data are provided in the Source data.

10) The authors should replace all gene and protein names and , locus tags used to identify the experimentally verified proteins with the locus tag present in dbVgrG. This is essential since the names used in the blotting and other analysis cannot be confidently identified without using dbVgrG.

We add the locus tag present in dbVgrG in revised Fig 3 and Fig 5 as suggested.

11) The relationship between the primers (Table S2) and their targets is not clear to me: I could not find even a subsequence of the primers, or their reverse complement, matching the genomes and genes they were supposed to amplify. Also, for the NlpC/P60 gene from *Burkholderia* sp. RPE67, the reverse primers sequence is missing.

As the Reviewer suggested, we have added comments at the bottom of Table S2 and detailed description in revised Materials and Methods as following: “Recombinant plasmids were constructed by pEASY®-Uni Seamless Cloning and Assembly Kit (TransGen Biotech). The tags were added by overlap extension PCR. Genes from *K.alkanivorans* GI5 and *Burkholderia* sp.RPE67 were synthesized and constructed to pETduet and RSFduet vectors by Biomed Biotech. DNA sequences and Primers used in this study were synthesized by RuiBiotech and listed in Table S2 (line 444-448).” The genes from *K.alkanivorans* GI5 and *Burkholderia* sp.RPE67 in this study are synthesized. Therefore, HindIII-R in vector was used as the reverse primer sequence for the NlpC/P60 domain from *Burkholderia* sp. RPE67. To avoid confusion, we have changed the name of

“HindIII-R” into “NlpC_P60-R”.

Reviewers' comments:

Reviewer #1 (Remarks to the Author):

In this revised version, Wang et al made some minor modifications to their manuscript. Mainly, they have dropped the concept of "co-effectors" and are trying to refer to the domains they found to be common in VgrG-containing operons as "conserved domains". They also provide additional computational analyses for LysM domains.

However, after the revisions, I am not sure that I understand what is the take-home message of the work. What do the authors actually claim is the role of the domains that they found, aside from these domains being present in various forms within VgrG-containing loci. In some places, it seems that the authors still claim these domains play a role in the delivery of effectors (title and abstract, for example). In other instances, they describe them as required for loading onto the tube-spike, and elsewhere they are simply described as "conserved domains".

Regarding the lack of experimental validation for the claims on the role of the LysM domain in effector secretion – I believe that the inability to obtain a strain does not enable one to maintain weak claims as they are. Under such circumstances, one should modify their claims and conclusions to include only what is firmly supported by the available results. The authors claim to have "toned down" their conclusions regarding LysM domains in this revised version. I don't think that they have (see my comments above). The manuscript still contains multiple claims of the domain required for effector delivery (see title and abstract line 39-40) without a single secretion or competition assay being provided.

Additional, specific concerns that remain:

- The revised title mentions "... revealed conserved domains for effector delivery...". The most important word in this title is missing, and it should come after "domains". What is it? If the authors still wish to claim these domains are required for effector delivery, they must provide direct experimental evidence to support that. This evidence is still lacking in the revised manuscript. Otherwise, are these simply domains often found within VgrG-containing loci. If so, then the manuscript should better explain what they might be doing and why they are found in the different forms within these loci. The authors should also perform wider analyses of these domains in an inclusive context to determine whether they are indeed "genetically linked" to the T6SS as they claim.

- Abstract:

Lines 36-38 (abstract): "... LysM... confirmed to be indispensable for... delivery...". There is still no experimental evidence confirming this claim.

Lines 40-41: As I mentioned before, LysM domains are not specifically associated with T6SS effectors in any way (they can be found in many eukaryotes), so they cannot be "distinct features" of T6SS effector recruitment. The authors now provide a phylogenetic tree of LysM domains (only in bacteria and phages), but they do not analyze potential differences that may differentiate between LysM domains that are encoded within T6SS cluster and those that do not (which could have helped their claims of LysM being somewhat linked to T6SS).

- Introduction:

Line 60: The change is incorrect. The domains only sometimes neighbor VgrG genes.

Lines 92-94: This sentence is supposed to be the bottom-line message concluded from this work, but I can't understand what it means or claims.

- Results:

Lines 98-99: As I mentioned before, only one example of a "MIX stand-alone gene" is known. In contrast to what is claimed in this sentence, this example is not part of a VgrG locus or operon.

Fig. S9 – When using AlphaFold predictions to support a hypothesis of protein-protein interactions, the predicted aligned error values generated by the prediction must be provided to determine whether the prediction and orientation of the two proteins/domains are well defined. Visualization of the PDB file alone is insufficient to draw any conclusions regarding a possible interaction between protein domains. Also, the PDB file generated by AlphaFold should be provided to allow reviewers and readers to assess the results.

- Discussion:

Line 344-346: Since the authors removed their claims of using the identified domains as markers for T6SS effectors, the connection of the first sentence to the paragraph is unclear.

Lines 348-351: This goes back to my comment on the original manuscript. How do the authors define "genetically linked"? The data in Table S1 are biased since they are derived only from VgrG loci. Therefore, the authors have no evidence to actually support a specific genetic link between the identified domains and VgrG.

- Other:

In their response to my previous comment #4, the authors claim that the presence of LysM and MIX in the same protein is uncommon. I believe this is incorrect. I refer the authors to the following paper on MIX domains (PMID: 30400344). Supplementary Dataset S2 provides a list of domains identified within MIX domain-containing proteins. Within the list, they will not only find multiple examples of proteins containing both MIX and LysM domains, but also examples of proteins containing MIX, LysM and another of their identified domains, PG_binding, all together in the same protein (for example, WP_061036948.1). In light of this, I think that the authors should provide a plausible explanation as to how one protein can have 3 domains relevant to interaction with VgrG/delivery/any other role they envision for the conserved domains.

Reviewer #1 (Remarks to the Author):

In this revised version, Wang et al made some minor modifications to their manuscript. Mainly, they have dropped the concept of “co-effectors” and are trying to refer to the domains they found to be common in VgrG-containing operons as “conserved domains”. They also provide additional computational analyses for LysM domains.

However, after the revisions, I am not sure that I understand what is the take-home message of the work. What do the authors actually claim is the role of the domains that they found, aside from these domains being present in various forms within VgrG-containing loci. In some places, it seems that the authors still claim these domains play a role in the delivery of effectors (title and abstract, for example). In other instances, they describe them as required for loading onto the tube-spike, and elsewhere they are simply described as “conserved domains”.

Regarding the lack of experimental validation for the claims on the role of the LysM domain in effector secretion – I believe that the inability to obtain a strain does not enable one to maintain weak claims as they are. Under such circumstances, one should modify their claims and conclusions to include only what is firmly supported by the available results. The authors claim to have “toned down” their conclusions regarding LysM domains in this revised version. I don’t think that they have (see my comments above). The manuscript still contains multiple claims of the domain required for effector delivery (see title and abstract line 39-40) without a single secretion or competition assay being provided.

Thank you for your comments. Although we explored the role of DUF2345 in the T6SS effector delivery, the function of LysM domain is not confirmed in a T6SS model (secretion or competition assay) as the appropriate bacterial strains are unavailable. We agreed with the Reviewer that it is inappropriate to claim all these conserved domains play a role in the delivery of effectors. We then modified the conclusions about LysM domains in the revised manuscript. Considering we showed that DUF2345 and LysM could bind both VgrG and toxin, to make this manuscript more rigorous, we modified the title as “Genome wide analysis revealed conserved domains involved in the effector discrimination of bacterial type VI secretion system”. Meanwhile, we also modified these claims throughout the manuscript for clarity (line 32-34; 37-39; 91-92; 270).

Additional, specific concerns that remain:

- The revised title mentions “... revealed conserved domains for effector delivery...”. The most important word in this title is missing, and it should come after “domains”. What is it? If the authors still wish to claim these domains are required for effector delivery, they must provide direct experimental evidence to support that. This evidence is still lacking in the revised manuscript. Otherwise, are these simply domains often found within *VgrG*-containing loci. If so, then the manuscript should better explain what they might be doing and why they are found in the different forms within these loci. The authors should also perform wider analyses of these domains in an inclusive context to determine whether they are indeed “genetically linked” to the T6SS as they claim.

In this manuscript, we showed two of these six conserved domains, DUF2345 and LysM, involved in the interaction between *VgrG* and effector, which is necessary for effector discrimination. As we could not provide direct experimental evidence to support that LysM domain is required for effector delivery, we modified the title as explained above. Except that, we found these conserved domains within *vgrG* loci are present in different encoding forms, a possible scenario could be the multiple origin and convergent evolution of these domains. We mentioned this issue in revised Discussion section. (line 346-350)

As Figure 1 showed, based on our *VgrG* database which include all currently available bacterial genomes, all these six conserved domains identified here are located between *vgrG* and toxin genes with the same transcriptional direction. Although it is reasonable to speculate that *vgrG*, conserved domain and toxin gene are within one locus, we deleted “genetically linked” to avoid misunderstanding. (line 350-353)

- Abstract:

Lines 36-38 (abstract): “... LysM... confirmed to be indispensable for... delivery...”. There is still no experimental evidence confirming this claim.

We modified this claim as the Reviewer suggested. (line 32-34)

Lines 40-41: As I mentioned before, LysM domains are not specifically associated with T6SS effectors in any way (they can be found in many eukaryotes), so they cannot be “distinct features”

of T6SS effector recruitment. The authors now provide a phylogenetic tree of LysM domains (only in bacteria and phages), but they do not analyze potential differences that may differentiate between LysM domains that are encoded within T6SS cluster and those that do not (which could have helped their claims of LysM being somewhat linked to T6SS).

As the Reviewer pointed out that LysM domains are not specifically associated with T6SS effectors, we deleted “distinct” and modified this sentence as “Together, our results implied that these widely distributed domain families with similar genetic configurations may be required for the T6SS effector recruitment process.”(line 37-39). Our phylogenetic tree showed that there are potential differences among VgrG, eCIS and phage-related LysM domains (Figure S10). In Discussion, we added “Considering LysM domains are not exclusively linked to T6SS-associated loci in bacterial genomes, it implies that some (if not all) of these domain-containing proteins might be multi-functional.” (line 310-312)

- Introduction:

Line 60: The change is incorrect. The domains only sometimes neighbor VgrG genes.

It is corrected as “These domains are sometimes encoded adjacent to *vgrG*...” (line 60)

Lines 92-94: This sentence is supposed to be the bottom-line message concluded from this work, but I can’t understand what it means or claims.

For clarity, we modified this sentence as “which are potentially important in the discrimination process of T6SS effectors” and deleted the last sentence of this paragraph. (line 91-93)

- Results:

Lines 98-99: As I mentioned before, only one example of a “MIX stand-alone gene” is known. In contrast to what is claimed in this sentence, this example is not part of a VgrG locus or operon.

Although 17 MIX stand-alone genes within *vgrG* loci were found in this study (Table S1), only one example was functional confirmed previously. We deleted “in the *vgrG* loci” for clarity. (line 97)

Fig. S9 – When using AlphaFold predictions to support a hypothesis of protein-protein interactions, the predicted aligned error values generated by the prediction must be provided to determine

whether the prediction and orientation of the two proteins/domains are well defined. Visualization of the PDB file alone is insufficient to draw any conclusions regarding a possible interaction between protein domains. Also, the PDB file generated by AlphaFold should be provided to allow reviewers and readers to assess the results.

The predicted aligned error values generated by the prediction was provided as following (Figure i). Apart from the amino acid residues spanning positions 500-600 and 650-700, where the structural uncertainty is relatively high, the predictions for residues in other regions are comparatively reliable. Also, the PDB file generated by AlphaFold is provided as supplementary data in revised manuscript.

Figure i: The predicted aligned error values generated by the prediction.

Discussion:

Line 344-346: Since the authors removed their claims of using the identified domains as markers for T6SS effectors, the connection of the first sentence to the paragraph is unclear.

We deleted this sentence. (line 342-344)

Lines 348-351: This goes back to my comment on the original manuscript. How do the authors define "genetically linked"? The data in Table S1 are biased since they are derived only from VgrG loci. Therefore, the authors have no evidence to actually support a specific genetic link between the identified domains and VgrG.

VgrG had been reported to be related to its downstream T6SS effectors (Durand et al., 2014;

Whitney et al., 2014). In this study, we used VgrG as a marker to identify potential T6SS related conserved domains. In Discussion section (line 354-359), we mentioned this bias that the data of this study are derived only from *vgrG* loci. As Figure 1 showed, these conserved domains are located between *vgrG* and toxin genes with the same transcriptional direction. Although it is reasonable to speculate that they are “genetically linked”, as the Reviewer suggested, we deleted this sentence to avoid misunderstanding. (line 350-353)

- Other:

In their response to my previous comment #4, the authors claim that the presence of LysM and MIX in the same protein is uncommon. I believe this is incorrect. I refer the authors to the following paper on MIX domains (PMID: 30400344). Supplementary Dataset S2 provides a list of domains identified within MIX domain-containing proteins. Within the list, they will not only find multiple examples of proteins containing both MIX and LysM domains, but also examples of proteins containing MIX, LysM and another of their identified domains, PG_binding, all together in the same protein (for example, WP_061036948.1). In light of this, I think that the authors should provide a plausible explanation as to how one protein can have 3 domains relevant to interaction with VgrG/delivery/any other role they envision for the conserved domains.

Thank you for reminding us there are more examples encoding more than one conserved domains in a single locus. Co-adaptor has been reported to improve the stability/expression of T6SS adaptor. Further, a recently paper reported that a secreted adaptor (RIX-containing protein) can enable the T6SS-mediated delivery of other cargo effectors by a previously undescribed mechanism (Kanarek et al., 2023). All these cases indicated that T6SS may employ some more sophisticated loading mechanism than expected. Although the potential function of more than one conserved domains encoded in one *vgrG* locus is interesting, it is not the main focus of this manuscript. We mentioned this issue in Discussion section in the submitted and major revision versions of the manuscript. As the Reviewer suggested, we further modified the description in the Discussion section and cited the paper (Dar et al., 2018; Kanarek et al., 2023). (line 319-335)

Reference:

Dar, Y., Salomon, D., and Bosis, E. (2018). The Antibacterial and Anti-Eukaryotic Type VI Secretion System MIX-Effector Repertoire in Vibrionaceae. *Mar Drugs* *16*.

Durand, E., Cambillau, C., Cascales, E., and Journet, L. (2014). VgrG, Tae, Tle, and beyond: the versatile arsenal of Type VI secretion effectors. *Trends Microbiol* *22*, 498-507.

Kanarek, K., Fridman, C.M., Bosis, E., and Salomon, D. (2023). The RIX domain defines a class of polymorphic T6SS effectors and secreted adaptors. *Nat Commun* *14*, 4983.

Whitney, J.C., Beck, C.M., Goo, Y.A., Russell, A.B., Harding, B.N., De Leon, J.A., Cunningham, D.A., Tran, B.Q., Low, D.A., Goodlett, D.R., *et al.* (2014). Genetically distinct pathways guide effector export through the type VI secretion system. *Mol Microbiol* *92*, 529-542.

REVIEWERS' COMMENTS:

Reviewer #4 (Remarks to the Author):

In this second revised version of the manuscript by Wang et al, the authors made some minor modifications to their manuscript according to the previous comments, they have modified their claims and conclusions, deleted some sentences, "to include only what were supported by the available results". In those regards, the authors have addressed the previous concerns.

Mainly, the domains they found to be common in VgrG-containing operons are described now as "conserved domains". The take-home message is thus they found conserved domains in various forms in VgrG-containing operons. Among them, DUF2345 are indispensable for interaction and T6SS effector delivery, while LysM domain can assist the interaction between VgrG and the corresponding toxin. The most complete results therefore concern DUF2345. As noted in the discussion, DUF2345 domains were already found to be important for interaction with toxin effector and delivery in *E. coli* (in a VgrG-fused context) (Flaunatti et al, Mol Microbiol 2016; Flaunatti et al 2020 EMBO J). The observation of the conservation of these domains in VgrG-containing operons, as suggested by Boyer et al 2009, alone or in fusion with VgrG is an interesting finding.

In particular, as pointed out by the authors, the DUF2345 is predicted to fold as a b-helix and thus would be an extension of gp5-C domain, as shown in the structure of *E. coli* VgrG (Flaunatti et al 2020). Contacts between this b-prism domain and the phospholipase toxin were identified, but other features were shown to be important for the interaction: the toxin also interacts with the long helices parallel to the b-prism (in the sequence, these two helices are found immediately upstream the DUF2345 domain). Another important domain for the interaction is the TTR domain, that follows in the sequence the DUF2345 (Flaunatti et al 2016; 2020). These three features (helices-DUF2345-TTR) would correspond to the "COG4253" domain (that thus includes DUF2345) (Flaunatti et al 2020). In the 3 examples chosen by the authors, these additional motifs seem to present as well: e.g., VgrG2b also possess the two long helices (a.a 561-581 and 592-602), DUF2345 and the b-sheet TTR domain (780-833). Of note, this TTR domain in VgrG2b was detected in Wood et al paper as well (Wood et al, Cell Rep 2019, Figure 1B). In the AlphaFold model of the complexes presented in Figure S5 of this manuscript, these features are indeed visible and seems to be important features for the predicted interaction with M35 domain. My question is thus the following: when the authors found conserved DUF2345 in vgrG or vgrG loci, are DUF2345 always associated with these features? Are COG4253 conserved, or is it just the DUF2345? Or other additional sequence can be found in addition to the DUF2345 b-helix? The presence of a TTR domain or any additional domain in addition to the DUF2345 b-helix could explain the specificity of interaction with different toxins (discussion line 225-226 "Thus, our data reveals that, DUF2345 domains exhibit high sequence diversity and likely interact with their cognate toxins specifically"). Please clarify this point as this information may help to go further in the analysis, and the additional/extensions in DUF2345 could explain the different specificities.

In the same line: Line 216 "DUF2345 can also interact with the toxin protein": and Figure 3G. Is DUF2345 a DUF2345 only? (just the b-helix portion of the protein?). From the primers table S2 (RSFduet-Flag-DUF-R), it seems that the pRSF-FLAG_DUF encodes the full length AKO63-2954, thus not only the DUF2345 domain, but the whole COG4253, that looks indeed from the AlphaFold model to include an additional domain (corresponding to a b-sheet domain TTR?) that interacts with the M35 domain in the model. There is thus no proof here that the DUF2345 per se is enough to interact with the M35 toxin. In this case, line 300-302 "Our results revealed that DUF2345 is required for the ultimate translocation of effector proteins by the interaction with both the VgrG protein and its cognate effector" should be corrected/downtuned.

REVIEWERS' COMMENTS:

Reviewer #4 (Remarks to the Author):

In this second revised version of the manuscript by Wang et al, the authors made some minor modifications to their manuscript according to the previous comments, they have modified their claims and conclusions, deleted some sentences, "to include only what were supported by the available results". In those regards, the authors have addressed the previous concerns.

Mainly, the domains they found to be common in VgrG-containing operons are described now as conserved domains. The take-home message is thus they found conserved domains in various forms in VgrG-containing operons. Among them, DUF2345 are indispensable for interaction and T6SS effector delivery, while LysM domain can assist the interaction between VgrG and the corresponding toxin. The most complete results therefore concern DUF2345. As noted in the discussion, DUF2345 domains were already found to be important for interaction with toxin effector and delivery in *E. coli* (in a VgrG-fused context) (Flaughnatti et al, Mol Microbiol 2016; Flaughnatti et al 2020 EMBO J). The observation of the conservation of these domains in VgrG-containing operons, as suggested by Boyer et al 2009, alone or in fusion with VgrG is an interesting finding.

In particular, as pointed out by the authors, the DUF2345 is predicted to fold as a β -helix and thus would be an extension of gp5-C domain, as shown in the structure of *E. coli* VgrG (Flaughnatti et al 2020). Contacts between this β -prism domain and the phospholipase toxin were identified, but other features were shown to be important for the interaction: the toxin also interacts with the long helices parallel to the β -prism (in the sequence, these two helices are found immediately upstream the DUF2345 domain). Another important domain for the interaction is the TTR domain, that follows in the sequence the DUF2345 (Flaughnatti et al 2016; 2020). These three features (helices-DUF2345-TTR) would correspond to the "COG4253" domain (that thus includes DUF2345) (Flaughnatti et al 2020). In the 3 examples chosen by the authors, these additional motifs seem to present as well: e.g., VgrG2b also possess the two long helices (a.a 561-581 and 592-602), DUF2345 and the β -sheet TTR domain (780-833). Of note, this TTR domain in VgrG2b was detected in Wood et al paper as well (Wood et al, Cell Rep 2019, Figure 1B). In the AlphaFold model of the complexes presented in Figure S5 of this manuscript, these features are indeed visible and seems to be important features for the predicted interaction with M35 domain. My question is thus the following: when the authors found conserved DUF2345 in vgrG or vgrG loci, are DUF2345 always associated with these features? Are COG4253 conserved, or is it just the DUF2345? Or other additional sequence can be found in addition to the DUF2345 β -helix? The presence of a TTR domain or any additional domain in addition to the DUF2345

b-helix could explain the specificity of interaction with different toxins (discussion line 225-226 "Thus, our data reveals that, DUF2345 domains exhibit high sequence diversity and likely interact with their cognate toxins specifically"). Please clarify this point as this information may help to go further in the analysis, and the additional/extensions in DUF2345 could explain the different specificities.

We appreciated the valuable comments raised by the Reviewer. In this manuscript, we scanned the conserved domains by batch CD-search from NCBI. The DUF2345 domain was identified, which is shorter than COG4253 and apparently does not include the upstream two helices and downstream TTR of COG4253 (Fig i). However, as shown in the Alphafold model (Figure S5), the DUF2345 domain in these three examples are associated with two helices and TTR. Thus, although these two regions (two helices and TTR) cannot be predicted by CDD, it is quite possible that most of DUF2345 are associated with these two regions.

We modified the manuscript as the reviewer suggested (line 223-224; line 299; line 305-306).

Fig i: Conserved domains of VgrG2b by CD-search from NCBI

In the same line: Line 216 "DUF2345 can also interact with the toxin protein": and Figure 3G. Is DUF2345 a DUF2345 only? (just the b-helix portion of the protein?). From the primers table S2 (RSFduet-Flag-DUF-R), it seems that the pRSF-FLAG_DUF encodes the full length AKO63-2954, thus not only the DUF2345 domain, but the whole COG4253, that looks indeed from the Alphafold model to include an additional domain (corresponding to a b-sheet domain TTR?) that interacts with the M35 domain in the model. There is thus no proof here that the DUF2345 per se is enough to interact with the M35 toxin. In this case, line 300-302 "Our results revealed that DUF2345 is required for the ultimate translocation of effector proteins by the interaction with both the VgrG protein and its cognate effector" should be corrected/downtoned.

As the Reviewer mentioned, the full length AKO63-2954 (TTR included) was used for IP experiments. Although DUF2345 domain (TTR not included) in VgrG2b was verified to assist the interaction between VgrG domain and M35 domain (Figure 3), to make the conclusions more rigorous, we toned down these descriptions (line 195-200; line 212-213; line 301-306).